# Continual Model Merging without Data: Dual Projections for Balancing Stability and Plasticity

**Enneng Yang**[1,2,5*]**, Anke Tang**[4]**, Li Shen**[1,3†]**, Guibing Guo**[2†]**, Xingwei Wang**[2]**, Xiaochun Cao**[1]**, Jie Zhang**[5]

[1] Shenzhen Campus of Sun Yat-sen University, China [2] Northeastern University, China
[3] Shenzhen Loop Area Institute, China [4] Wuhan University, China [5] Nanyang Technological University, Singapore
ennengyang@gmail.com; anketang@whu.edu.cn; shenli6@mail.sysu.edu.cn; guogb@swc.neu.edu.cn
wangxw@swc.neu.edu.cn; caoxiaochun@mail.sysu.edu.cn; zhangj@ntu.edu.sg

## Abstract

Model merging integrates multiple expert models with diverse capabilities into a unified framework, facilitating collaborative learning. However, most existing methods assume simultaneous access to all models, which is often impractical in real-world scenarios where models are received sequentially. While some studies have investigated continual model merging (CMM)–which involves sequentially merging multiple models–the challenge of balancing prior knowledge (stability) and incorporating new tasks (plasticity) remains unresolved. This paper, for the first time, formally defines the stability and plasticity of CMM from the perspective of orthogonal projection. Subsequently, we analyze the relationships among the spaces spanned by task data, historical gradients, and accumulated gradients. Building on this, we propose a data-free **D**ual **O**rthogonal **P**rojection (DOP) method, which eliminates data dependence and mitigates interference between the merged model and models for old and new tasks by projecting their parameter differences onto their respective approximate data spaces. Finally, to solve potential conflicts between stability and plasticity, we reformulate DOP as a multi-objective optimization problem and employ a multi-gradient descent algorithm to obtain a Pareto-optimal solution. Extensive experiments across multiple architectures and task configurations validate that our approach significantly outperforms state-of-the-art CMM methods.

## 1 Introduction

Fine-tuning pre-trained foundation models for task- or domain-specific applications has become a dominant paradigm in machine learning [50, 102]. The open-source culture in the machine learning community has further accelerated this trend, driving the rapid expansion of fine-tuned expert models. For instance, Huggingface [84] currently provides access to over 1.6 million models. A key challenge is how to effectively leverage these diverse expert models to address novel or more complex tasks. Recently, model merging has gained prominence as a promising solution by integrating multiple expert models with identical architectures at the parameter level. As a general technique, it has diverse practical applications, including multi-task learning [35, 98, 2], continual pre-training/fine-tuning [64, 86, 106, 52], zero/few-shot learning [92, 58, 34, 32], model attack/protection [101, 6] and others [30, 9, 32].

Traditional model merging settings primarily aim to resolve parameter conflicts during the merging process [93, 103]. To address this challenge, researchers have introduced various innovative strategies, including weighted-based merging methods [53, 37, 97], subspace-based approaches [98, 91],

---

[*] This work was completed during Enneng Yang's study and employment at institutions $1, 2, 5$.
[†] Corresponding authors.

39th Conference on Neural Information Processing Systems (NeurIPS 2025).

and dynamic merging techniques [90, 51]. However, these approaches are built on the fundamental assumption that all models are *simultaneously available* for merging. In practical scenarios, tasks or domains typically emerge incrementally, and preserving past data or models indefinitely is often infeasible [22, 81, 104, 79]. Furthermore, loading all expert models into memory simultaneously during merging becomes infeasible when the number of tasks is large or the models are of substantial size, resulting in excessive memory consumption. To address these challenges, two recent studies have introduced the concept of *continual/temporal* model merging (CMM) [24, 74], which aims to merge a sequence of models arriving incrementally. Among them, TIME [24] is an analytical study that highlights a key conclusion: Due to the greater complexity of CMM setting, traditional merging methods struggle to generalize effectively to this setting. OPCM [74] introduced a method tailored for the CMM setting, employing parameter decomposition to mitigate catastrophic forgetting [63] of previous models when merging new ones. However, OPCM fails to account for the interference that prior tasks may introduce to newly learned tasks, ultimately limiting overall performance.

In this paper, we are the first to formally define two key properties that CMM should satisfy, namely *stability* and *plasticity*, from the perspective of orthogonal projection [100, 67], thereby aligning them with the objectives of traditional continual learning [79, 104, 81]. Stability [63] ensures that the continually merged model retains knowledge of previously merged tasks, while plasticity [22] enables the model to effectively integrate knowledge from new tasks. However, as the original task data is unavailable in the CMM setting, existing orthogonal projection techniques [67, 49, 95] in continual learning cannot be employed. In addition, we demonstrate across vision and language tasks that existing CMM methods struggle to effectively balance these two objectives (in §3.2.1). To address this limitation, we analyze the relationships among the subspaces spanned by data, gradients, and accumulated gradients (in §4.1). Our analysis reveals that in the linear model, the accumulated gradient subspace is contained within the gradient subspace, which in turn lies within the data subspace. Therefore, the space spanned by the accumulated gradients can be viewed as an approximation of the space spanned by the data.

Building on these findings, we propose a data-free **D**ual **O**rthogonal **P**rojection (DOP) method that minimizes the projections of parameter differences between the merged model and both the old and new models in the approximate data subspace (in §4.2). Furthermore, to achieve a better trade-off between stability and plasticity in CMM, we reformulate DOP as a multi-objective optimization problem and employ the multi-gradient descent algorithm (MGDA) [20] to obtain a Pareto-optimal solution. Additionally, we incorporate an exponential moving average strategy to reduce fluctuations during the MGDA optimization process. It is worth noting that our approach is both memory- and time-efficient, requiring at most only three linear layers to be loaded into memory during the model merging optimization process; furthermore, the average optimization time for a single linear layer is only 1 to 3 seconds. Finally, we conducted extensive experiments across three vision architectures, one language architecture, and three different task quantities (i.e., 8/14/20). The results demonstrate that our method achieves the best performance under the CMM setting by better balancing stability and plasticity. The contributions can be summarized into the following:

- We formalize the two key optimization objectives of CMM, namely stability and plasticity, within the framework of orthogonal projection theory, and we further examine the relationships among the subspaces spanned by task data, gradients, and accumulated gradients.
- We propose a data-free Dual Orthogonal Projection (DOP) method for CMM, recasting it as a multi-objective optimization problem to obtain a Pareto-optimal solution, ensuring an effective trade-off between stability and plasticity.
- We conduct extensive CMM experiments on four architectures (covering vision and language models) with varying task numbers, and show that our method consistently outperforms state-of-the-art (SOTA) CMM approaches on both old and new tasks.

## 2   Related Work

This paper is related to model merging [68, 35, 92, 2] and continual learning [104, 79, 81]. Due to space limitations, we provide a brief overview in this section and discuss the details in Appendix B.

**Traditional Model Merging**: Based on the merging strategies, existing methods can be roughly divided into three main categories [93, 103]: (1) *Weight-based Methods*: These methods employ grid search, heuristic strategies, or learnable approaches to determine the optimal merging coefficients,

thereby mitigating task interference [85, 35, 105, 97, 53, 37]. (2) *Subspace-based Methods*: These approaches project the full parameter space into an appropriate subspace and then merge the parameters within this subspace to reduce task conflicts [91, 29, 98, 78, 33, 106, 25, 83]. (3) *Router-based Methods*: These methods dynamically adjust model parameters for each task or instance during inference, rather than using a shared parameter set across all tasks or instances [90, 73, 51]. However, the aforementioned methods have a fundamental limitation: they necessitate *simultaneous* access to all models (i.e., "offline" merging), thereby limiting their applicability in streaming scenarios.

**Temporal and Continual Model Merging**: To the best of our knowledge, only two studies have explored the CMM setting [24, 74], in which sequentially arriving models are merged into a unified model. TIME [24] introduced the concept of temporal model merging and examined the impact of various merging techniques, initialization strategies, and deployment approaches. It concluded that standard offline merging techniques generalize poorly to CMM, underscoring the challenges of this setting. However, TIME primarily functions as an analytical pipeline for evaluating existing methods, without introducing a tailored approach for CMM. The study most closely related to this work is OPCM [74], which mitigates interference with prior tasks by decomposing the parameters of new models in order to address catastrophic forgetting. However, OPCM primarily addresses the forgetting induced by the new model on the old model, while overlooking the influence of the old model on the new model, resulting in suboptimal performance.

# 3 Preliminaries

## 3.1 Notations and Problem Definitions

Consider a scenario with $T$ expert models, denoted as $\{\Theta_1, \Theta_2, \ldots, \Theta_T\}$. Each model $\Theta_t$ is independently fine-tuned from a common pre-trained model $\Theta_0$ (e.g., ViT [23, 61]) on a domain- or task-specific dataset $\mathcal{D}_t^{tr} = \{X_t^{tr}, Y_t^{tr}\}$, where $X_t^{tr}$ and $Y_t^{tr}$ denote the input data and corresponding labels. The aim of model merging is to merge expert models at the *parameter level* to construct a unified model $\Theta_*$ that preserves the capabilities of all experts without requiring access to their respective training data. Formally: $\Theta_* = \texttt{merge}(\Theta_1, \Theta_2, \ldots, \Theta_T)$, where $\texttt{merge}()$ is a merging operation [85, 53, 37]. A straightforward approach is weight averaging [85]: $\Theta_* = \frac{1}{T}\sum_{t=1}^{T}\Theta_t$. Recent studies indicate that task vector-based model merging [35, 57, 91, 97] frequently surpasses simple weight averaging in performance. Specifically, the task vector [35] for task $t$ is defined as the difference between the fine-tuned parameters $\Theta_t$ and the pre-trained parameters $\Theta_0$, i.e., $\tau_t = \Theta_t - \Theta_0$. Based on task vectors, the model merging problem is redefined as Definition 3.1.

**Definition 3.1** (*Task Vector-based Model Merging*). Given a pre-trained model $\Theta_0$ and multiple expert task vectors $\{\tau_1, \tau_2, \ldots, \tau_T\}$, the goal is to aggregate the pre-trained model and expert models to produce a unified model $\Theta_*$, defined as: $\Theta_* = \texttt{merge}(\Theta_0, \tau_1, \tau_2, \ldots, \tau_T)$.

However, traditional model merging faces a key limitation: it requires all expert models to be available *simultaneously*, which is often unrealistic. To overcome this challenge, we aim to adapt model merging to an incremental setting, referred to as Continual Model Merging (CMM) [74]. The definition of CMM is provided in Definition 3.2.

**Definition 3.2** (*Continual Model Merging*). Given a pre-trained model $\Theta_0$ and a sequence of models $\{\Theta_1, \Theta_2, \ldots, \Theta_T\}$ arriving sequentially, the goal is to incrementally merge these models at the parameter level. At each time step $t$, the merged model $\Theta_*^{(t)}$ integrates information from the previously merged model $\Theta_*^{(t-1)}$ and the newly arrived task model $\Theta_t$ (or task vector $\tau_t$), ensuring high performance across all tasks (i.e., $1, 2, \ldots, t$) without requiring access to the original training data or previously accessed expert models. The process of CMM is formalized as: $\Theta_*^{(t)} = \texttt{merge}(\Theta_0, \Theta_*^{(t-1)}, \tau_t)$, where $t \geq 2$.

As task-specific expert models cannot be accessed simultaneously, the CMM setting poses a greater challenge than the traditional model merging setting. Prior studies have demonstrated that existing model merging methods have failed to generalize effectively in the CMM setting [24, 74], and our results in the next section further support this conclusion.

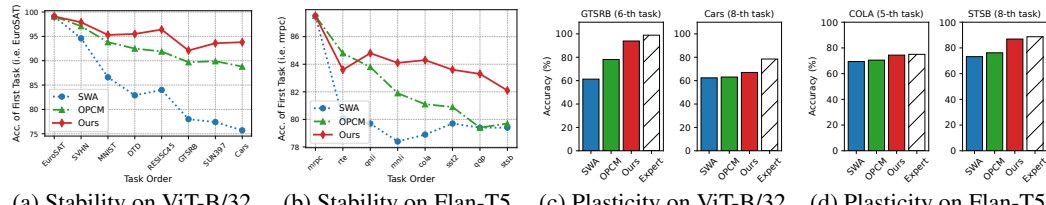

| (a) Stability on ViT-B/32 | (b) Stability on Flan-T5 | (c) Plasticity on ViT-B/32 | (d) Plasticity on Flan-T5 |

Figure 1: Analysis of different CMM methods for stability of old tasks (a-b) and plasticity of new tasks (c-d). This analysis covers the vision model ViT-B/32 and the language model Flan-T5-Base.

## 3.2 Rethinking Stability and Plasticity in CMM

Unlike traditional model merging, which primarily aims to mitigate task interference [91, 93], we argue that CMM should simultaneously satisfy two key properties–stability and plasticity–in alignment with the principles of traditional continual learning [81, 104, 79]: (1) **Stability**: Preventing catastrophic forgetting of previously merged tasks when incorporating new models [63]. (2) **Plasticity**: Enabling the model to effectively assimilate knowledge from new models [22].

### 3.2.1 The Challenge of Stability and Plasticity in CMM

This section first analyzes the performance of existing CMM methods in terms of stability and plasticity. Without loss of generality, we take the most classic method, SWA [36], and the SOTA method, OPCM [74], as representative methods. Details of both methods are in Appendix D. Experiments are conducted on two architectures: ViT-B/32 [23, 61] and Flan-T5-Base [14].

The results for **stability** are shown in Fig.1(a) and Fig.1(b). We evaluate the performance of the merged model on the first task after sequentially merging up to the $t$-th model ($t \in \{1, 2, \ldots, 8\}$) according to task order. We observed a clear performance degradation for both SWA and OPCM, indicating poor stability. For instance, after merging eight models using SWA and OPCM, the performance of ViT on the first task (i.e., EuroSAT) dropped from 99.1 to 75.7 and 88.8, respectively. Similarly, on Flan-T5, the performance on the first task (i.e., MRPC) drops from 87.5 to 79.4 and 79.7. This suggests that current CMM methods exhibit limited stability.

The results for **plasticity** are shown in Fig.1(c) and Fig.1(d). We evaluate the models performance on the newly merged task at step $t$. A score closer to the expert model (i.e., a model fine-tuned independently on that task) indicates better plasticity. We find that on the 8-th task (i.e., Cars), the expert model on the ViT architecture achieved a score of 78.5, while SWA and OPCM only reached 62.4 and 63.1, respectively. On Flan-T5, for the 8th task (i.e., STSB), the expert model achieved 88.7, while SWA and OPCM achieved only 73.2 and 76.2, respectively. These results demonstrate that current CMM methods also perform poorly regarding plasticity.

### 3.2.2 Reinterpreting Stability and Plasticity from the Perspective of Orthogonal Projection

In this section, we analyze how an ideal CMM satisfies stability and plasticity from the perspective of orthogonal projection [67]. Recall Definition 3.2 of CMM: at time step $t$, the available models include the pre-trained model $\Theta_0$, the previously merged model $\Theta_*^{(t-1)}$, and the newly arrived model $\Theta_t$ (or task vector $\tau_t$). In particular, from the perspective of network architecture and parameter distribution, Transformer-based architectures [75, 23] (Fig. 4(a) in the Appendix) allocate most of their parameters to linear transformations, including the Q/K/V projection layers in multi-head attention, the two linear transformations in the feed-forward network, and the embedding layers. Our analysis in Fig. 4(b) in the Appendix indicates that linear layer parameters constitute $90 - 99\%$ of the entire model (i.e., $\Theta_0$). Therefore, effective linear layer merging is crucial for successful model merging [37]. Next, we illustrate how to decompose the model merging problem into a sequence of linear layer merging sub-problems, incorporating stability and plasticity.

Consider two linear layers, $f(X_o) = W_o^\top X_o$ and $f(X_n) = W_n^\top X_n$, where $W_o, W_n \in \mathbb{R}^{d_1 \times d_2}$ denote the weight matrices of a linear layer in old $\Theta_*^{(t-1)}$ and new $\Theta_t$ models, respectively. $X_o, X_n \in \mathbb{R}^{d_1}$ represent the input data for the old and new tasks, while $d_1$ and $d_2$ denote the input and output dimensions of the linear layer. To achieve model merging, the goal is to integrate the linear layer parameters $W_o$ and $W_n$ to produce a new linear layer with parameters

$W_* = \texttt{merge}(W_o, W_n)$. Alternatively, this can be expressed in the form of task vector-based merging: $W_* = \texttt{merge}(W_0, \tau_o, \tau_n)$, where $\tau_o = W_o - W_0$ and $\tau_n = W_n - W_0$ represent the task vectors, i.e., parameter shifts from a shared pre-trained model $W_0$, for the old and new models, respectively.

To minimize the loss of the merged linear layer $W_*$ across both the old and new tasks, we seek to simultaneously satisfy the following stability and plasticity objectives:

$$\min_{W_*} \{\mathcal{L}_s, \mathcal{L}_p\} \text{ s.t. } \mathcal{L}_s = d(W_*^\top X_o - W_o^\top X_o), \quad \mathcal{L}_p = d(W_*^\top X_n - W_n^\top X_n), \tag{1}$$

where $d(\cdot, \cdot)$ denotes an arbitrary distance function, such as $L_2$ loss. To further refine the formulation, we introduce the delta parameters of the merged model relative to the old model $W_o$ and the new model $W_n$ as follows: $\Delta W_o^* = W_* - W_o, \Delta W_n^* = W_* - W_n$. Thus, $\mathcal{L}_s$ and $\mathcal{L}_p$ can be equivalently reformulated as $\mathcal{L}_s = d(\Delta W_o^* X_o)$ and $\mathcal{L}_p = d(\Delta W_n^* X_n)$.

Furthermore, based on the properties of orthogonal projection theory [67, 49, 95] in continual learning, if $\Delta W_o^*$ is orthogonal to the subspace spanned by the old task's data $X_o$, i.e., $\Delta W_o^* \perp \texttt{span}(X_o)$, then $\Delta W_o^* X_o = 0$, allowing us to deduce that: $W_* X_o = (W_o + \Delta W_o^*) X_o = W_o X_o$. In other words, the merged linear model $W_*$ produces exactly the same output as the expert model $W_o$ for the old task, thereby preserving the capabilities of the old task. Similarly, if the projection of the delta parameters $\Delta W_n^*$ onto the input space spanned by the new task's data $X_n$ is 0 (i.e., lies in the orthogonal complement: $\Delta W_n^* \perp \texttt{span}(X_n)$), then $W_* X_n = (W_n + \Delta W_n^*) X_n = W_n X_n$. That is, the merged $W_*$ effectively incorporates the knowledge of the new task, meaning that it produces the same response as $W_n$ to the $X_n$.

Building upon the definition of the delta parameters and the theory of orthogonal projection, the two objectives of CMM–stability and plasticity–can be reformulated as *minimizing the projections of the delta parameters onto their respective input spaces*. However, in the CMM setting, the data $X_o$ and $X_n$ for the previous and new tasks are *unavailable*, and the spaces spanned by these data cannot be directly obtained. Instead, only the expert models trained on these tasks are available, posing a significant challenge to solving CMM.

# 4 Methodology

This section first analyzes the correlation among the subspaces spanned by data, gradients, and task vectors in §4.1. Leveraging this correlation, we introduce a data-free DOP method for CMM in §4.2.

## 4.1 Subspace Correlation Analysis

*Property* 4.1 (***The Gradient Space within the Input Space***). For a linear layer $f(X) = W^\top X$, the space spanned by the gradients $\nabla_W \mathcal{L}$ with respect to its weight matrix $W$ lies within the space spanned by the input data $X$.

*Proof.* Consider a linear layer with input $X$, ground truth $Y$, weight matrix $W \in \mathbb{R}^{d_1 \times d_2}$, and prediction $Z = f(X) = \sigma(W^\top X)$, where $\sigma(\cdot)$ is a nonlinear activation function, and $d_1$ and $d_2$ represent the input and output dimensions of the linear layer, respectively. Let the loss function be $\mathcal{L}(Z, Y)$, which is differentiable with respect to $f(X)$ and can take various forms (e.g., cross-entropy, L2 loss, or other differentiable functions). By applying the chain rule, we obtain: $\frac{\partial \mathcal{L}}{\partial W} = \frac{\partial \mathcal{L}}{\partial Z} \frac{\partial Z}{\partial W} = \left(\frac{\partial \mathcal{L}}{\partial Z} \odot \sigma'\right) X^\top = [a_1 X, a_2 X, \ldots, a_{d_2} X]$, where $\odot$ denotes element-wise multiplication, and $[a_1, a_2, \ldots, a_{d_2}]$ refers to the vector formed by $\frac{\partial \mathcal{L}}{\partial Z} \odot \sigma'$. It clearly shows that each column of the gradient $\nabla_W \mathcal{L}$ is a scalar multiple of the input $X$, i.e., $\alpha_k$ ($k \in \{1, 2, \ldots, d_2\}$). This result confirms that the gradient $\nabla_W \mathcal{L}$ resides within the subspace spanned by the data $X$ [67, 49, 95, 48, 12].

*Property* 4.2 (***The Task Vector Space within the Gradient Space***). Within the gradient-based iterative update framework, given the initial parameters $W_0$, the learning rates $\{\eta_{t,i}\}_{i=1}^k$, and the gradients $\{\nabla_{W_{t,i}} \mathcal{L}\}_{i=1}^k$ from $k$ updates, the task vector $\tau_t$ is directly associated with the accumulated gradients. Specifically, the task vector is computed as the weighted sum of the gradients, where the weights correspond to the learning rates across all training steps. Consequently, the task vector $\tau_t$ resides within the subspace spanned by these historical gradients.

*Proof.* Consider a task undergoing $k$ iterative updates during fine-tuning (e.g., via SGD or its variants [65]). According to the standard gradient descent update rule, the parameter updates are given

by: $W_{t,k} = W_{t,k-1} - \eta_{t,k} \cdot \nabla_{W_{t,k}} \mathcal{L} = W_0 - \sum_{i=1}^{k} \eta_{t,i} \cdot \nabla_{W_{t,i}} \mathcal{L}$, Consequently, the difference between the parameters $W_{t,k}$ at the step $k$ and the initial $W_0$ corresponds to the cumulative sum of the updates at each step [89, 83, 12]. This is consistent with the definition of the task vector [35], which is $\tau_t = W_{t,k} - W_0 = -\sum \eta_{t,i} \cdot \nabla_{W_{t,i}} \mathcal{L}$.

## 4.2 Data-free Dual Orthogonal Projection Method

In the optimization objective of CMM (in §3.2.2), the unavailability of original data $X_o$ and $X_n$ makes it infeasible to construct input subspaces $\mathrm{span}(X_o)$ and $\mathrm{span}(X_n)$. Motivated by Properties 4.1 and 4.2, we propose approximating the input subspaces using the subspaces spanned by the task vectors $\tau_o$ and $\tau_n$, i.e., $\mathrm{span}(\tau_o)$ and $\mathrm{span}(\tau_n)$. Notably, for the newly arrived model, two properties hold strictly. For the old merged model, the task vector, gradient, and data can be interpreted as the combined (or approximated) result of multiple tasks. The CMM objective is thus reformulated as minimizing the projections of the delta parameters $\Delta W_o^* = W_* - W_o$ and $\Delta W_n^* = W_* - W_n$ onto the task vector subspaces: $\min_{W_*} \{\mathcal{L}_s, \mathcal{L}_p\}$, where $\mathcal{L}_s = \mathrm{Proj}\left(\Delta W_o^*, \mathrm{span}\left(\tau_o\right)\right), \mathcal{L}_p = \mathrm{Proj}\left(\Delta W_n^*, \mathrm{span}\left(\tau_n\right)\right)$. Although $\tau_o$ and $\tau_n$ are commonly referred to as "task vectors" [35], they are in fact matrices of the same shape as $W_o$ and $W_n$, i.e., $\mathbb{R}^{d_1 \times d_2}$.

To construct the subspaces spanned by task vectors $\tau_o$ and $\tau_n$, we apply singular value decomposition (SVD) [41] to each as follows: $U_o\,\Sigma_o\,V_o^\top \approx \mathrm{SVD}_r(\tau_o), U_n\,\Sigma_n\,V_n^\top \approx \mathrm{SVD}_r(\tau_n)$, where the columns of $U_o$ and $V_o$ form an orthonormal basis for the row space and column space of $\tau_o$, respectively. In practice, we can retain only the top $r < \min(d_1, d_2)$ singular vectors and singular values, rather than employing the full decomposition. To capture both row-wise and column-wise directions, we leverage the left and right singular matrices obtained from the SVD decomposition. Additionally, to penalize projections along principal directions, we retain the singular values $\Sigma_o$ or $\Sigma_n$ as weighting factors to regulate projection strength along each singular direction. By minimizing the projections of $\Delta W_o^*$ and $\Delta W_n^*$ onto the subspaces spanned by $\tau_o$ and $\tau_n$, respectively, such that their Frobenius norms are reduced, we ensure that $W_*$ retains prior task knowledge, thereby maintaining stability. Simultaneously, this approach enables $W_*$ to effectively incorporate knowledge from new tasks, thereby preserving plasticity. Since this optimization does not require access to the original data, we introduce the *data-free **D**ual **O**rthogonal **P**rojection* (DOP) objective:

$$\min_{W_*}\{\mathcal{L}_s, \mathcal{L}_p\} \text{ s.t. } \mathcal{L}_s = ||U_o^\top \Sigma_o \Delta W_o^*||_F^2 + ||\Delta W_o^* \Sigma_o V_o||_F^2, \mathcal{L}_p = ||U_n^\top \Sigma_n \Delta W_n^*||_F^2 + ||\Delta W_n^* \Sigma_n V_n||_F^2. \quad (2)$$

A straightforward approach is to combine the two loss terms via weighted summation and optimize $W_*$ iteratively (e.g., using SGD or Adam [65]): $\mathcal{L}_{CMM} = \alpha \mathcal{L}_s + (1 - \alpha)\mathcal{L}_p$, where $\alpha$ serves as a trade-off parameter between stability and plasticity. However, grid search for $\alpha$ is computationally expensive, and a fixed weighting scheme often fails to achieve an optimal trade-off due to the inherent conflict between stability and plasticity.

We formulate DOP as a multi-objective optimization problem to balance these competing objectives and seek the Pareto-optimal solution, thereby ensuring both stability and plasticity. Without losing generality, we employ the multi-gradient descent algorithm (MGDA) [20] to iteratively solve this multi-objective optimization problem, that is $\min_{W_*}\{\mathcal{L}_s, \mathcal{L}_p\}$ in §4.2. The update rule of the merged linear model $W_*$ at each step $k$ is given by: $W_{*,k+1} = W_{*,k} - \eta \cdot g_k$, where $W_{*,0} = (W_o + W_n)/2$, $\eta$ is a global learning rate like SGD or adaptive learning rate as Adam [39] at step $k$, and $g_k$ is a gradient update direction. $W_{*,0}$ is initialized using simple weight averaging [85], which can be easily replaced by existing model merging methods. More specifically, MGDA seeks to determine an updated direction $g_k$ at each iteration $k$ to simultaneously minimize all objective functions. This is formulated as the following quadratic programming problem, and the one-dimensional optimization problem for $\alpha_k$ admits an analytical solution:

$$\min_{\alpha_k \in [0,1]} \left\| \alpha_k \cdot \nabla_{W_{*,k}} \mathcal{L}_s + (1 - \alpha_k) \cdot \nabla_{W_{*,k}} \mathcal{L}_p \right\|_2^2, \text{ s.t. } \alpha_k = \mathtt{clamp}\left( \frac{\left(\nabla_{W_{*,k}} \mathcal{L}_p - \nabla_{W_{*,k}} \mathcal{L}_s\right)^\top \nabla_{W_{*,k}} \mathcal{L}_p}{\left\|\nabla_{W_{*,k}} \mathcal{L}_s - \nabla_{W_{*,k}} \mathcal{L}_p\right\|_2^2} \right),$$
$$(3)$$

where the function $\mathtt{clamp}(x) = \max(\min(x, 1), 0)$ ensures $\alpha_k$ remains within $[0, 1]$, thereby maintaining a valid convex combination of the two gradients.

To prevent abrupt changes in $\alpha_k$ at each update step $k$ and to incorporate a preference $\gamma \in (0, 1)$ (e.g., prioritizing stability or plasticity), we adopt an exponential moving average (EMA) strategy, a commonly used approach in adaptive learning rate optimizers [65]. Specifically, given $\alpha_k$ computed

Table 1: Performance comparison of continual (C.) merging methods on 8/14/20 visual tasks, reporting ACC, and BWT over ten randomly shuffled task orders. Ref. is the abbreviation of reference.

| | Method | ViT-B/32 | | | ViT-B/16 | | | ViT-L/14 | | |
|---|---|---|---|---|---|---|---|---|---|---|
| | | 8 tasks | 14 tasks | 20 tasks | 8 tasks | 14 tasks | 20 tasks | 8 tasks | 14 tasks | 20 tasks |
| Ref.(↑) | Pre-Trained | 48.1 | 56.9 | 55.6 | 55.4 | 62.0 | 59.8 | 64.9 | 69.1 | 65.6 |
| | Single Fine-Tuned | 90.4 | 89.3 | 89.8 | 92.4 | 91.3 | 91.6 | 94.3 | 93.4 | 93.5 |
| | C. Fine-Tuned | 79.8 | 67.4 | 62.6 | 82.9 | 72.2 | 68.2 | 90.0 | 70.9 | 77.7 |
| | Continual Model Merging Methods | | | | | | | | | |
| ACC (↑) | Averaging (SWA) | 66.3±0.0 | 65.4±0.0 | 61.1±0.0 | 72.3±0.0 | 69.7±0.0 | 64.8±0.0 | 80.0±0.0 | 77.5±0.0 | 71.1±0.0 |
| | C. Task Arithmetic | 67.5±0.0 | 66.5±0.0 | 60.6±0.0 | 77.1±0.0 | 70.9±0.0 | 64.2±0.0 | 82.1±0.0 | 77.9±0.0 | 70.3±0.0 |
| | C. Ties-Merging | 49.0±10.2 | 66.2±0.6 | 59.9±0.7 | 66.8±3.7 | 70.5±0.8 | 63.0±1.6 | 64.3±7.0 | 78.0±0.6 | 68.3±0.9 |
| | OPCM | 75.5±0.5 | 71.9±0.3 | 65.7±0.2 | 81.8±0.3 | **77.1±0.5** | 70.3±0.2 | 87.0±0.4 | 83.5±0.2 | 76.0±0.2 |
| | **DOP (Ours)** | **78.3±1.6** | **73.0±1.2** | **68.5±1.6** | **84.5±0.8** | 76.2±1.2 | **72.2±1.8** | **88.3±0.7** | **85.5±0.8** | **80.7±0.9** |
| BWT (↑) | Averaging (SWA) | -11.5±2.2 | -8.0±1.3 | -7.1±2.1 | -9.7±1.5 | -7.1±1.4 | -7.3±1.7 | -7.3±1.4 | -5.8±1.0 | -6.4±1.5 |
| | C. Task Arithmetic | -9.6±1.5 | -1.3±0.3 | -3.4±0.4 | -4.2±1.0 | -1.3±0.4 | -3.6±0.4 | -7.1±0.8 | -1.8±0.3 | -3.3±0.3 |
| | C. Ties-Merging | -15.3±8.0 | 1.9±0.6 | -1.5±0.7 | -5.5±0.4 | 1.4±0.7 | -1.5±1.2 | -13.0±5.7 | 1.1±0.4 | -2.9±1.0 |
| | OPCM | -6.3±1.1 | -6.0±1.0 | -7.8±1.5 | -4.8±0.7 | -5.1±1.4 | -6.3±2.2 | -2.6±1.0 | -4.3±0.7 | -6.5±1.8 |
| | DOP (Ours) | -6.1±2.3 | -8.8±2.1 | -12.5±2.8 | -3.9±1.0 | -7.5±2.2 | -11.4±2.8 | -3.2±1.2 | -4.6±1.0 | -8.4±1.4 |

at step $k$, we apply the following smoothing strategy: $\alpha_{s,k} = \beta\alpha_{s,k} + (1-\beta)\alpha_k, \alpha_{p,k} = \beta\alpha_{p,k} + (1-\beta)(1-\alpha_k)$, where $\alpha_{s,0} = \gamma, \alpha_{p,0} = 1-\gamma$, and the smoothing factor $\beta \in [0,1)$ controls the degree of historical information incorporated into the current update, while $\gamma$ determining the initial preference for stability or plasticity. Finally, the optimal update direction $g_k$ at iteration $k$ is given by: $g_k = \alpha_{s,k}\nabla_{W_{*,k}}\mathcal{L}_s + \alpha_{p,k}\nabla_{W_{*,k}}\mathcal{L}_p$. This update process is repeated iteratively until convergence. We give the pseudocode of the DOP algorithm in Appendix E.

# 5 Experiment

## 5.1 Experimental Setup

**Models and Datasets**: For *visual* tasks, our experimental setup follows OPCM [74]. We adopt three ViT models of varying scales from CLIP [61] as backbones: ViT-B/32, ViT-B/16, ViT-L/14. Their statistical details are presented in Fig. 4. We fine-tune three architectures on 20 datasets to obtain task-specific expert models for continual merging. For each architecture, we consider three task number settings: 8 tasks, 14 tasks, and 20 tasks. For *language* tasks, following FusionBench [72], we primarily use the Flan-T5-Base [14] architecture and fine-tune it on the eight GLUE [77] tasks to obtain eight expert models for continual merging. Dataset details are provided in Appendix C.

**Baselines**: We primarily compare four state-of-the-art model merging methods: Averaging (SWA) [30, 85], C. Task Arithmetic [35], C. Ties-Merging [91], and OPCM [74]. Detailed descriptions of these baselines are provided in Appendix D. We also include the pre-trained model, individual expert models, and continually fine-tuned models as reference baselines.

**Evaluation Metrics**: We mainly measured the final merged model's average task accuracy (ACC) across all evaluation tasks, where a higher ACC indicates better retention of old and new task capabilities. We also evaluate Backward Transfer (BWT), where a higher BWT, meaningful only under comparable accuracy, indicates less forgetting.

## 5.2 Evaluating the Performance of Continuously Merged Visual Models

**Accuracy**: As shown in Tab. 1, we make the following key observations: (1) The pre-trained model lacks task-specific knowledge, resulting in poor performance. Single fine-tuned models achieve the best performance but necessitate maintaining and deploying separate model parameters for each task. Continual fine-tuning allows a single model to sequentially learn downstream tasks; however, due to catastrophic forgetting, its performance falls between that of the pre-trained and fine-tuned models. As the number of tasks increases, the performance of continual fine-tuning deteriorates significantly, approaching that of the pre-trained model. (2) Among all CMM methods, OPCM exhibits superior performance compared to SWA, C. Task Arithmetic, and C. Ties-Merging. This is attributed to OPCM being specifically designed for the CMM setting, leveraging parameter decomposition to mitigate forgetting of old tasks. In contrast, the other methods were originally designed for settings where all models are available simultaneously, making them less effective in the CMM [24]. (3) Our DOP method consistently demonstrates superior performance across nearly all scenarios, owing

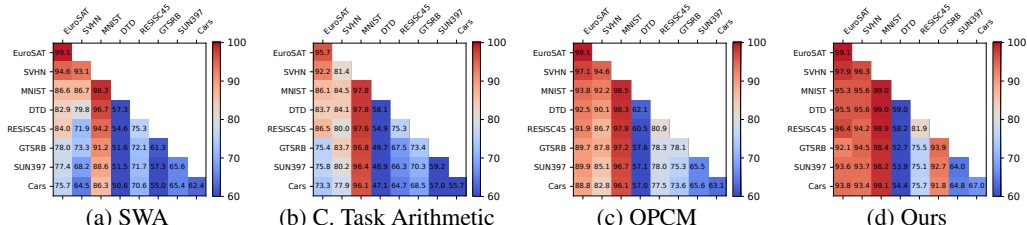

Figure 2: Performance during merging of 8 tasks with ViT-B/32. Row $t$ shows accuracy on previous tasks $(1-t)$ after merging task $t$ (stability); diagonal shows accuracy on task $t$ itself (plasticity).

Table 2: Experimental results of continually merging Flan-T5-base models on all eight tasks.

| Method | CoLA | MNLI | MRPC | QNLI | QQP | RTE | SST2 | STSB | ACC(↑) | BWT(↑) |
|---|---|---|---|---|---|---|---|---|---|---|
| (Ref.) Single Fine-Tune | 69.1 | 82.7 | 85.5 | 90.9 | 84.0 | 84.4 | 92.9 | 87.4 | 84.6±0.0 | - |
| Averaging (SWA) | 69.1 | 62.6 | 79.4 | 89.8 | **83.9** | 81.2 | 91.7 | 73.2 | 78.8±0.0 | -2.5±1.1 |
| C. Task Arithmetic | 70.5 | 57.8 | 78.4 | 90.2 | 83.6 | 80.5 | 92.3 | 77.8 | 78.9±0.0 | -1.8±1.1 |
| C. Ties-Merging | 68.8 | 50.5 | 79.8 | 89.9 | 83.1 | 79.2 | 91.9 | 79.3 | 77.8±1.6 | -0.5±1.9 |
| OPCM | 69.7 | 72.9 | 78.8 | 90.2 | 83.8 | **82.2** | 92.3 | 74.7 | 80.6±0.3 | -2.1±0.6 |
| DOP (Ours) | **72.6** | **80.9** | **81.0** | **90.4** | 83.4 | 81.0 | **92.9** | **86.1** | **83.6±0.1** | -1.3±0.3 |

to its effective balance of plasticity for new tasks and stability for old tasks via multi-objective optimization. For example, when merging 20 tasks using the ViT-L/14 model, our method attains an accuracy of 80.7, substantially surpassing the strongest baseline OPCM, which achieves 76.0.

**Stability and Plasticity**: While the BWT metric in Tab. 1 partially quantifies the degree of forgetting, it is only meaningful when comparing methods with comparable performance. When comparing DOP and OPCM, their BWT values exhibit slight differences but remain generally close. Moreover, in the CMM setting, the performance differences between methods are more pronounced, and task order can also influence the final results [74]. To more precisely evaluate the stability and plasticity of each method, we fix the task order and assess their performance on each task at different stages. As illustrated in Fig. 2, under a fixed task order, DOP exhibits significantly superior plasticity on new tasks. For example, when learning GTSRB (6-th row), DOP achieves an accuracy of 93.9, whereas OPCM, C. Task Arithmetic, and SWA achieve only 78.1, 73.4, and 61.3, respectively. Additionally, after merging all eight tasks (8-th row), DOP maintains a performance of 93.8 on the first task (EuroSAT), while the other three methods achieve only 88.8, 73.3, and 75.7. This suggests that DOP provides superior stability, effectively mitigating forgetting of old tasks. In addition, the results in Figs. 1(a,c) also indicate that our method has better stability and plasticity.

## 5.3 Evaluating the Performance of Continuously Merged Language Models

**Accuracy**: As shown in Tab. 2, we evaluate the performance of five CMM methods when merging eight expert models trained on the GLUE dataset. We observe that: (1) SWA, C. Task Arithmetic, and C. Ties-Merging achieve comparable performance, with scores of 78.8 and 78.9, respectively. Among them, C. Ties-Merging performs the worst, likely due to significant knowledge loss caused by its sparsification of task vectors during the merging process. (2) OPCM, the only method among the baselines specifically designed for CMM, achieves the second-best performance, with an average score of 80.6 across the eight tasks. (3) Most notably, our proposed DOP method significantly outperforms OPCM, reaching a score of 83.6, remarkably close to the performance of independently fine-tuned expert models, which achieve 84.6. While expert models deliver the best results, they require maintaining a separate model for each task, which is costly.

**Stability and Plasticity**: As shown in the BWT of Tab. 2, our DOP method demonstrates a small degree of forgetting at a higher average performance. In addition, as shown in Figs. 1(b,d), after merging all eight tasks, our method achieves a performance of 82.1 on the first task (i.e., MRPC), significantly outperforming OPCM and SWA, which achieve 79.7 and 79.4, respectively, demonstrating the superior stability of our approach. Furthermore, when merging the 8-th task (i.e., STSB), our method achieves an accuracy of 86.9, substantially higher than OPCMs 76.2 and SWAs 73.2, demonstrating greater plasticity of our DOP method.

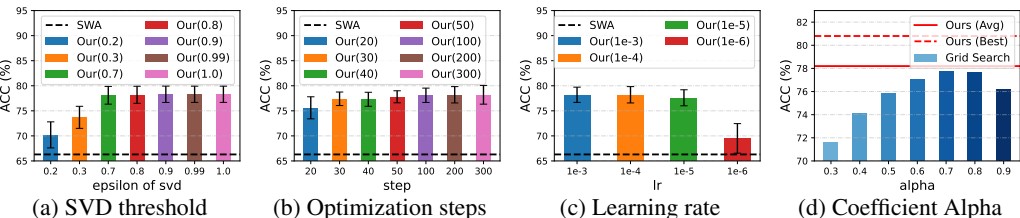

| (a) SVD threshold | (b) Optimization steps | (c) Learning rate | (d) Coefficient Alpha |

Figure 3: Hyperparameter effects on ViT-B/32 task merging (8 tasks).

## 5.4 Efficiency and Ablation Analysis

**Efficiency**: DOP employs the MGDA to iteratively refine the merged model. Tab. 3 presents the time required to optimize a single linear layer across four architectures. Our method demonstrates high computational efficiency, completing each layers optimization in just $1-3$ seconds, without requiring access to any original task data. As outlined

Table 3: Running time statistics

| Models | ViT-B/32 | ViT-B/16 |
|--------|----------|----------|
| Time (s) | 1.74±1.18 | 1.55±0.47 |
| Models | ViT-L/14 | Flan-T5-Base |
| Time (s) | 3.11±1.73 | 1.30±1.32 |

in the methodology, DOP primarily targets linear layers and optimizes stability and plasticity losses at the layer level. During merging only three linear layers need to be loaded: the pre-trained, the previously merged, and the new. The above results indicate that our method achieves high efficiency in both computation and memory usage.

**Multi-objective Optimization**: As shown in Tab. 4, to assess the robustness of multi-objective optimization, we replaced MGDA with a naïve approach that directly sums the two losses in Eq. 2. As a result, performance declined from 78.2 to 75.9, confirming the effectiveness of identifying the Pareto solution. The results in Fig. 3(d) fur-

Table 4: Analysis of MOO strategy

| Methods | MGDA | EMA | ACC (↑) | Std (↓) |
|---------|------|-----|---------|---------|
| w/o MGDA | ✗ | ✗ | 75.9 | 3.88 |
| w/o EMA | ✓ | ✗ | 76.3 | 3.66 |
| DOP | ✓ | ✓ | 78.2 | 1.63 |

ther illustrate that even with an extensive grid search to balance stability and plasticity losses, the performance remains inferior to the multi-objective optimization approach proposed in this paper. Moreover, removing the EMA strategy for loss weights in MGDA resulted in a further performance decline, underscoring the benefits of stable optimization introduced by EMA in Eq. 3.

## 5.5 Hyper-parameters Analysis

**SVD Threshold**: In the SVD-based subspace construction (in §4.2), only a subset of principal directions is retained for projection. As shown in Fig. 3(a), we experimented with different subspace retention ratios and observed that when the retention ratio is low (e.g., 0.2 or 0.3), the approximated subspace undergoes information loss, resulting in degraded performance. As the ratio increases, performance improves significantly. When the ratio exceeds 0.7, the overall performance stabilizes.

**Optimization Step**: In DOP optimization, we iteratively update the merged model parameters using MGDA (in §4.2). We evaluated the effect of varying iteration counts on performance. As depicted in Fig. 3(b), performance improves as the number of iterations increases. When the iteration count surpasses 50, performance stabilizes, demonstrating the high efficiency of our optimization approach.

**Learning Rate**: We analyzed the effect of different learning rates on performance (in §4.2). As shown in Fig. 3(c), when the learning rate is too small (e.g., $1e-6$), it fails to effectively optimize the stability and plasticity objectives, thereby limiting DOP's performance improvement. However, with appropriate learning rates (e.g., $1e-5$ to $1e-3$), DOP exhibits stable and consistent performance.

## 6 Conclusion

In this paper, we discuss the challenging CMM setting. We first formally define two key objectives for CMM–plasticity and stability–from the perspective of orthogonal projection. Next, we analyze the relationships among the subspaces spanned by task data, gradients, and task vectors. Building on these insights, we propose a data-free Dual Orthogonal Projection (DOP) method for CMM. By reformulating DOP as a multi-objective optimization problem, our method aims to find Pareto-optimal solutions that effectively balance stability and plasticity. Extensive experiments across three architectures and diverse task configurations validate the effectiveness and efficiency of our DOP.

## Acknowledgements

This research is supported by A*STAR under its RIE2025 Manufacturing, Trade and Connectivity (MTC) Industry Alignment Fund- Pre-Positioning (IAF-PP) (Award M23L4a0001), and by the National Natural Science Foundation of China under Grant (No. 62576083, 62032013, 62025604, 62576364), Shenzhen Basic Research Project (Natural Science Foundation) Basic Research Key Project (NO. JCYJ20241202124430041), the Open Research Fund from Guangdong Laboratory of Artificial Intelligence and Digital Economy (SZ) (No. GML-KF-24-23).

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

**Appendix Overview**. The content of this appendix is organized as follows:

- In Appendix A, we discuss the limitations and broader impact of this work.
- In Appendix B, we discuss the research progress on model merging and continual learning in detail.
- In Appendix C, we provide details on the dataset configurations and implementation of our experiment.
- In Appendix D, we present the core formulas used by existing baselines when performing continual model merging.
- In Appendix E, we provide the complete pseudocode of the proposed data-free dual orthogonal projection (DOP) method.
- In Appendix F, we present additional experimental results and analyses that could not be included in the main text due to space constraints.

## A  Limitations and Broader Impact

**Limitations**. Although this work demonstrates promising stability and plasticity under the continual model merging setting, there remain several limitations and potential directions for future exploration. First, this paper focuses on the most extreme setting of continual model merging, where no data is available. Future work could investigate the impact of incorporating a small amount of data during continual merging. Second, we primarily evaluate the proposed DOP method on vision and language models, lacking validation on larger-scale models and data of other modalities. In the future, we aim to extend continual model merging to large-scale multimodal settings.

**Broader Impact**. In this paper, we propose a novel approach for continual model merging. It holds the potential to enable continual accumulation of new knowledge without forgetting previous knowledge–an ability that serves as a cornerstone toward more advanced artificial intelligence, even in the absence of data. Moreover, this continual model merging setup naturally supports decentralized development: each user can locally train their own expert model, and these models can then be sequentially merged into a unified model without sharing data. As such, continual model merging offers an orthogonal and complementary alternative to traditional continual learning.

## B  Related Work

In this work, our research objective is to explore how to merge a sequence of expert models without forgetting the knowledge accumulated from previously merged models. This topic is related to both traditional model merging and continual learning, yet it is fundamentally distinct from both. In this section, we provide a detailed discussion on these distinctions.

### B.1  Model Merging

Model merging aims to integrate multiple models with diverse capabilities into a unified model at the "parameter level" [103, 93, 71, 26, 27, 85, 35, 2]. Based on the merging strategy, existing methods can be broadly categorized into static model merging and dynamic model merging. The former can be further divided into weight-based merging and subspace-based merging approaches.

Weight-based merging methods aim to assign appropriate merging coefficients to balance the contributions of different expert models to the final merged model [105, 97]. Using average weighting is the most straightforward and simple approach [30], but its performance is limited, especially when the expert models differ significantly. Task Arithmetic [35] adopts grid search, while Evolutionary model merge [2] and LoRAHub [32] employ evolutionary algorithms to search for optimal merging coefficients. However, such approaches become inefficient when the number of models is large. Some studies have proposed assigning merging coefficients based on statistical information or learnable strategies. For example, Fisher-Merging [53] uses the Fisher Information Matrix to measure parameter importance and assigns weights accordingly. RegMean [37] derives a closed-form solution for merging linear layers based on training data. AdaMerging [97] introduces an unsupervised loss based on entropy minimization to optimize layer-wise merging coefficients.

Subspace-based merging methods project models into a sparse or low-rank subspace to alleviate parameter interference during merging. Ties-Merging [91] removes the less significant components

of task vectors and aligns parameter signs before merging. DARE [98] randomly drops a proportion of elements in the task vectors and rescales the remaining parameters. TALL-masks [78] maintains a sparse mask matrix for each task to isolate conflicting parameters. Model Tailor [106] measures parameter importance based on loss sensitivity and removes unimportant parameters, though it relies on data availability. PCB-Merging [29] carefully evaluates the importance of parameters both within and across tasks to prune unimportant ones. TSV-M [25], STAR [46], SFTM [60], and AdaRank [45] perform singular value decomposition (SVD) on the weight matrices and merge expert models only within the low-rank subspace spanned by the most significant singular directions.

Dynamic merging methods differ from the approaches above in that they construct different merged models for each individual sample or task, thereby increasing the flexibility of model merging. For example, WEMoE [73] dynamically combines the multilayer perceptron modules of expert models through a learnable routing mechanism. Twin-Merging [51] and PHATGOOSE [54] dynamically compose a series of low-rank expert modules. DAWIN [56] merges the parameters of two models based on their predicted probabilities. However, due to the need to perform merging before each inference, such dynamic approaches typically incur significant latency during prediction.

Some other model merging methods operate during the fine-tuning stage by disentangling weights [57], aligning parameters before merging [1], or correcting representation bias after merging [94, 82]. In particular, our work shares some conceptual relevance with WUDI-Merging [12], as both leverage the correlation between task vectors and data in linear layers. However, there are key differences in the problem setting. WUDI-Merging assumes that all models are available simultaneously and focuses on resolving task conflicts. In contrast, our method assumes a sequential arrival of models and emphasizes balancing the stability-plasticity trade-off between new and old tasks. We also propose a multi-objective optimization strategy to dynamically adjust the combination coefficients for the two objectives.

**Continual Model Merging (CMM).** In summary, the aforementioned model merging methods all assume that *expert models are available simultaneously*, overlooking the practical scenario where expert models may arrive sequentially. While TIME [24] and OPCM [74] consider this setting, TIME is primarily an evaluation-focused work, and OPCM does not explicitly address or tailor its design for the stability-plasticity trade-off, which limits its performance in continual merging scenarios. In contrast, this paper is the first to formulate stability and plasticity as dual objectives in the continual model merging setting from the perspective of orthogonal projection, and proposes a targeted data-free optimization scheme, effectively filling this gap in the literature.

**Application Scenarios of CMM.** The practical scenarios for continual model merging include, but are not limited to: (i) *Memory-efficient model merging*: Traditional model merging requires loading all candidate models simultaneously, which is often infeasible in resource-constrained environments. CMM provides a lower-cost and more scalable alternative. (ii) *Continual model reuse*: When users wish to further enhance a composite model published by others (as in the HuggingFace example) by merging their own private models, CMM enables continual capability enhancement without requiring access to the original private models of previous users. (iii) *Improved model generalization*: Standard training often leads to convergence in sharp regions of the loss landscape, resulting in poor generalization. Continual merging of checkpoints can guide models toward flatter minima, thereby improving generalization [9, 30].

## B.2 Continual Learning

The goal of continual learning is to enable a deep neural network to continuously acquire knowledge from new tasks without forgetting the knowledge learned from previous tasks, thereby achieving a human-like ability for lifelong learning [104, 79, 81]. To mitigate catastrophic forgetting of old tasks, existing approaches can be broadly categorized into replay-based methods, architecture-based methods, regularization-based methods, and orthogonal projection methods.

Replay-based methods alleviate forgetting by replaying a small buffer of old task data while learning new tasks. The buffer can be filled by randomly sampling from previous data or by using various heuristic strategies. For example, iCaRL [62] selects samples closest to the class centers for replay, while RWalk [10] and MIR [4] select samples near the decision boundaries. Rainbow [7] emphasizes maximizing the diversity of replayed samples. Some other approaches use generative models to synthesize data from previous tasks for replay [13].

Architecture-based methods learn new tasks in sparse subspaces or progressively expand the network by adding new parameter modules to accommodate incoming tasks. For example, WSN [38] selects a winning subnetwork for each specific task based on the lottery ticket hypothesis. PNN [66] creates a separate branch for each task, while ExpertGate [5] adds a new expert module for every new task.

Regularization-based methods adopt the idea of knowledge distillation, treating the model trained on new tasks as the student and the model from previous tasks as the teacher, to supervise the student and prevent forgetting. LwF [47] uses the teacher models output as an additional training signal for the student model, while LwM [21] leverages attention maps from previous tasks as supervision. EWC [40] and MAS [3], on the other hand, constrain the parameter changes between the new model and the previous model to preserve prior knowledge.

Orthogonal projection-based methods update gradients in directions orthogonal to the subspace occupied by previous tasks, thereby preserving prior knowledge during new task learning [80]. For example, OWM [100] constructs the subspace of previous tasks based on historical gradients, while GPM [67] and DFGP [95, 96] build the old task subspace using feature representations from historical data across spaces. FS-DGPM [19] and TRGP [49] further refine this idea by scaling each basis vector in the old task subspace according to its importance.

However, all the aforementioned continual learning methods are *data-driven*, meaning they rely on continuously training the model on data from each task to obtain a unified model. As such, they are not applicable to our setting. In this work, we obtain a unified model by sequentially merging a series of expert models [24, 74], without directly accessing the original training data of each task. This novel approach (model-driven) offers an orthogonal perspective to traditional continual learning paradigms (data-driven).

## C  Experiment Details

**Vision Tasks**: For image classification tasks, we fine-tuned three architectures on 20 datasets, consistent with OPCM [74], to obtain task-specific expert models for merging. For each architecture (Clip-ViT-B/32, Clip-ViT-B/16 or Clip-ViT-L/14 [61, 23]), we consider three different task configurations: *(1) 8 Tasks*: SUN397 [88], Cars [42], RESISC45 [11], EuroSAT [31], SVHN [99], GTSRB [70], MNIST [44], DTD [15]. *(2) 14 Tasks*: In addition to the 8 tasks above, the following 6 tasks are included: Flowers102 [55], PCAM [76], FER2013 [28], OxfordIIITPet [59], STL10 [17], CIFAR100 [43]. *(3) 20 Tasks*: Building on the 14 tasks above, the following 6 tasks are added: CIFAR10 [43], Food101 [8], FashionMNIST [87], EMNIST [18], KMNIST [16], and RenderedSST2 [69, 61]. These datasets cover a diverse range of task types, including digit recognition, satellite image classification, and natural image classification, among others. We fine-tuned using cross-entropy loss and Adam with cosine annealing (max learning rate: 1e-5, batch size: 128) for 4,000 steps as Fusionbench [72].

**Language Tasks**: For natural language processing tasks, we select 8 text-to-text generation tasks from the GLUE benchmark, including CoLA, MNLI, MRPC, QNLI, QQP, RTE, SST-2, and STSB. Following the prompt settings in FusionBench [72], we fine-tune the Flan-T5-Base [14] model on each task to build expert models. During evaluation, we report Spearmans $\rho$ for the STSB task, and exact match accuracy for the other seven tasks. Specifically, for the STSB task, if the textual output cannot be parsed into a numerical value, we set $\rho$ to 0.

**Implementation Details**: For all baseline methods, we employed the same hyperparameter search strategy as OPCM [74]. For our proposed method, we utilized the Adam optimizer with an initial learning rate of $1e-4$ and set the number of iterations to 200 to optimize the DOP objectives. The initial value of the weight in MGDA is set to 0.8. For the EMA smoothing strategy, we set $\beta$ to 0.999. All vision model experiments were conducted on a single NVIDIA RTX A6000 GPU (48 GB), while all language model experiments were conducted on a single NVIDIA V100 GPU (32 GB). All results are averaged over ten runs, and both the mean and standard deviation are reported.

## D  Existing CMM Methods

To the best of our knowledge, nearly all existing model merging methods are designed for multi-task model merging (i.e., assuming simultaneous availability of all models), with only a few studies

addressing CMM [24, 74]. Among the comparison methods, TIME [24] functions as an evaluation pipeline, whereas OPCM constitutes a state-of-the-art CMM algorithm [74]. To facilitate a comprehensive comparison with our approach, we further adapted Stochastic Weight Averaging (SWA) [36], Task Arithmetic [35], and Ties-Merging [91] to the continual model merging setting as additional baselines for comparison. The core operations of these methods are summarized as follows.

***Stochastic Weight Averaging (SWA)*** [36, 85]:

$$\Theta_*^{(t)} = \frac{1}{2}\Theta_*^{(t-1)} + \frac{1}{2}\Theta_t, t \geq 2, \tag{4}$$

***Continual Task Arithmetic*** [35]:

$$\Theta_*^{(t)} = \Theta_0 + \lambda \cdot (\tau_{\Theta_*^{(t-1)}} + \tau_t), t \geq 2, \tag{5}$$

where $\tau_{\Theta_*^{(t-1)}} = \Theta_*^{(t-1)} - \Theta_0$, that is, the task vector constructed by the merged model of step $t-1$ and the pre-trained model. $\lambda$ is a hyperparameter to control the importance of the task vectors when merging them with the pre-trained model.

***Continual Ties-Merging*** [91]:

$$\Theta_*^{(t)} = \Theta_0 + \lambda \cdot (\hat{\tau}_{\Theta_*^{(t-1)}} + \hat{\tau}_t), t \geq 2, \tag{6}$$

where $\hat{\tau}_{\Theta_*^{(t-1)}}$ and $\hat{\tau}_t$ represent the task vectors $\tau_{\Theta_*^{(t-1)}}$ and $\tau_t$ after resolving sign conflicts using the "Trim" and "Elect Sign" strategies from original Ties-Merging [91].

***Orthogonal Projection-based Continual Merging (OPCM)*** [74]:

$$\Theta_*^{(t)} = \Theta_0 + \frac{\lambda^{(t-1)} \cdot \tau_{\Theta_*^{(t-1)}} + \mathcal{P}^{(t-1)}(\tau_t)}{\lambda^{(t)}}, t \geq 2, \tag{7}$$

where $\mathcal{P}^{(t-1)}(\tau_t)$ is a projection operation that projects $\tau_t$ to the orthogonal directions of the task vector spanning the subspace at step $t-1$.

# E    Pseudo-code of the DOP Algorithm

As shown in Alg. 1, we provide the complete pseudocode for the DOP method. The merging process in our algorithm consists of two main components: DOP optimization for linear layers (lines 3-19) and a simple parameter averaging strategy for non-linear layers (lines 21-23). This step can optionally be replaced with any existing model merging strategy. It is worth noting that the optimization of linear layers can be executed in *parallel* across all layers.

More specifically, the core steps of DOP are as follows: (1) Lines 3-7: Perform SVD on the task vectors to construct the core subspaces. (2) Line 8: Use simple parameter averaging to initialize the merged model. (3) Lines 10-19: Optimize the merged linear layers, which involves computing the stability and plasticity losses (lines 12-13), calculating the gradient descent direction based on multi-objective optimization (lines 14-17), and updating the parameters via gradient descent (line 18). Specifically, `getW()` and `setW()` denote the operations for retrieving and updating the layer weights, respectively.

# F    Experimental Results and Analysis

**Architecture and Parameter Distribution Analysis:** The Transformer [75] architecture has emerged as one of the most widely adopted frameworks in machine learning, demonstrating remarkable performance in both vision [23] and natural language processing tasks [75]. Transformer-based models are constructed by stacking multiple Transformer blocks. Fig. 4 (a) illustrates the standard architecture of a Transformer block, comprising multi-head attention layers, normalization layers, and feed-forward layers. Notably, the fundamental components of the multi-head attention and feed-forward layers are linear layers. As shown in Fig. 4 (b), linear layers constitute the majority of the models parameters. For example, in Flan-T5-Base, ViT-B/32, ViT-B/16, and ViT-L/14, linear layers account for 90.01%, 97.21%, 99.09%, and 99.68% of the total parameters, respectively. Thus, the key to merging Transformer-based models lies in effectively integrating their linear layers.

**Algorithm 1:** Dual Orthogonal Projection for Continual Model Merging

**Input:** pre-trained model $\Theta_0$, previously merged model $\Theta_{old}$, new model $\Theta_{new}$, hyper-parameters: $\gamma$, $K$, $r$, $\beta$, $\eta$

**Output:** merged model $\Theta_*$

```
1  for layer l in Θ₀ or Θ_old or Θ_new do
      // Linear layer optimization can be performed in parallel across
         all layers.
2     if layer l is a Linear layer then
         // Construct subspace using SVD
3        W₀, W_old, W_new ← getW(Θ₀, Θ_old, Θ_new, l);
4        τ_old ← W_old − W₀;
5        τ_new ← W_new − W₀;
6        U_o Σ_o V_o^⊤ ← svd_r(τ_old);
7        U_n Σ_n V_n^⊤ ← svd_r(τ_new);
         // Initialize the merged parameters and iterate the optimization
8        W_{*,0} ← (W_old + W_new)/2;
9        for step k in {1, 2, ..., K} do
            // Optimize stability and plasticity
10          ΔW*_old ← W_{*,k} − W_old;
11          ΔW*_new ← W_{*,k} − W_new;
12          L_s ← L_stability(ΔW*_old, U_o, Σ_o, V_o);
13          L_p ← L_plasticity(ΔW*_new, U_n, Σ_n, V_n);
            // Calculate the multi-objective balance coefficient
14          α_k ← clip( (∇_{W_{*,k}} L_p − ∇_{W_{*,k}} L_s)^⊤ ∇_{W_{*,k}} L_p / ‖∇_{W_{*,k}} L_s − ∇_{W_{*,k}} L_p‖₂² );
15          α_{s,k} ← β · α_{s,k−1} + (1 − β) · α_k;
16          α_{p,k} ← β · α_{p,k−1} + (1 − β) · (1 − α_k);
            // Calculate the gradient descent direction and perform the
               update
17          g_k ← α_{s,k} · ∇_{W_{*,k}} L_s + α_{p,k} · ∇_{W_{*,k}} L_p;
18          W_{*,k+1} ← W_{*,k} − η · g_k;
19       setW(Θ_*, l) ← W_{*,K};
20    else
         // Perform simple parameter averaging for nonlinear layers
21       W_old, W_new ← getW(Θ_old, Θ_new, l);
22       W_{*,0} ← (W_old + W_new)/2;
23       setW(Θ_*, l) ← W_{*,0};
24 return Θ_*;
```

**Stability and Plasticity Analysis**: In this section, we analyze how the performance of different CMM methods evolved over time under fixed model arrival orders. Fig. 5 and Fig. 6 illustrate the results of merging *vision tasks* under two specific task orders. Specifically, the task order in Fig. 5 is MNIST-SUN397-EuroSAT-RESISC45-DTD-SVHN-Cars-GTSRB, while the order in Fig. 6 is DTD-SVHN-GTSRB-MNIST-SUN397-EuroSAT-RESISC45-Cars. We observe that the proposed DOP method consistently demonstrates stronger plasticity and stability under both task orders. For example, in the first task order, when merging the eighth model, DOP achieved a performance of 68.0, whereas OPCM, C. Task Arithmetic, and SWA achieved only 63.9, 55.7, and 62.4, respectively–indicating superior plasticity of our method. At the same time, DOP maintains a performance of 94.1 on the second task, compared to 84.0, 77.9, and 64.5 for OPCM, C. Task Arithmetic, and SWA, respectivelydemonstrating better stability (i.e., less forgetting). Similarly, we also evaluate two random task arrival orders for the *language tasks*. Fig. 7 corresponds to the order: MRPC-RTE-QNLI-MNLI-CoLA-SST2-QQP-STSB, and Fig. 8 corresponds to: STSB-QQP-RTE-SST2-CoLA-QNLI-MRPC-MNLI. Consistently, we observe that our proposed method demonstrates superior plasticity and stability under both task orders.

**Single-task Performance Analysis**: For *visual classification tasks*, Fig. 9 and Fig. 10 depict the performance of the merged model on individual tasks when sequentially merging 8, 14, and 20 tasks using the ViT-B/32 and ViT-L/14 architectures. The performance of each task is normalized relative to its single fine-tuned model, with values closer to 1 indicating better performance. These

**(a) Transformer Encoder**

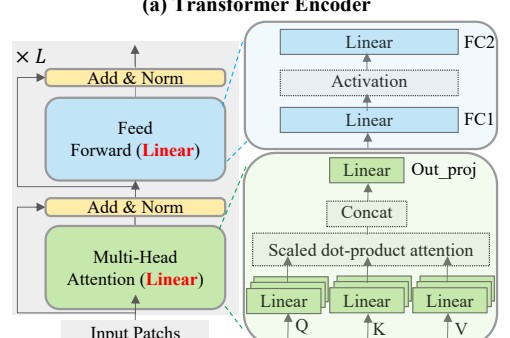

**(b) Number and ratio statistics of parameters**

| Architecture | ViT-B/32 | ViT-B/16 |
|---|---|---|
| Total Parameters | 87,455,232 | 85,798,656 |
| Linear Parameters | 85,017,600 | 85,017,600 |
| Linear Ratio | 97.21% | 99.09% |

| Architecture | ViT-L/14 | Flan-T5-Base |
|---|---|---|
| Total Parameters | 303,178,752 | 247,577,856 |
| Linear Parameters | 302,211,072 | 222,855,168 |
| Linear Ratio | 99.68% | 90.01% |

Figure 4: The total number of parameters and the statistics of linear layer parameters for three ViT architectures. (a) is a general Transformer block [75], including multi-head attention, feed-forward layers, layer normalization, etc. The primary components within the Transformer block are linear modules. (b) presents the statistics on the total parameters, linear layer parameters, and the parameters of linear layer weights (excluding bias terms) for the three vision architectures (i.e., ViT-B/32, ViT-B/16, ViT-L/14, and Flan-T5-Base).

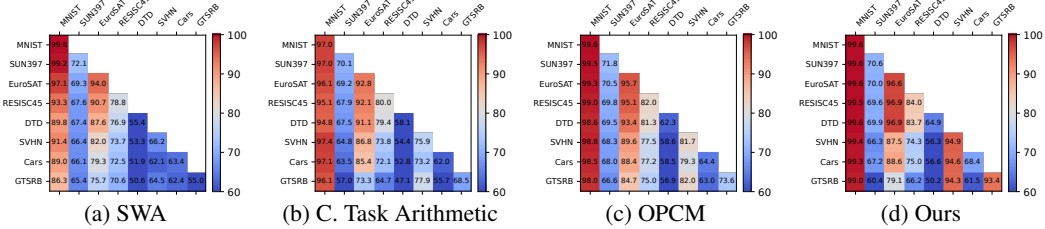

| (a) SWA | (b) C. Task Arithmetic | (c) OPCM | (d) Ours |
|---|---|---|---|

Figure 5: Performance changes during the merging of 8 tasks with the ViT-B/32 architecture. Row $t$ shows test accuracy on all previous tasks $(1 - t)$ after merging the $t$-th model (stability), while diagonal values indicate the model's plasticity, reflecting its performance on the newly merged task.

figures clearly show that the proposed DOP method consistently appears on the outermost ring, indicating superior performance on individual tasks compared to other CMM methods. For ***natural language processing tasks***, Table 2 presents the performance of various model merging methods as well as independently fine-tuned expert models. We observe that the proposed DOP method achieves performance on individual tasks that is closer to the expert models compared to other continual model merging approaches.

**Projection Space Analysis**: As detailed in §4.2, we approximate the input subspace using the subspace spanned by task vectors. In practice, this subspace is obtained via SVD decomposition of the task vectors. We analyze the performance variations when employing only the row space ($U$), only the column space ($V$), or both spaces for projection. As presented in Tab. 5, utilizing both subspaces preserves more spatial information and yields the best performance. In contrast, employing only the row or column space leads to suboptimal performance.

Table 5: Analysis of subspace projection strategy

| Methods | Space $V$ | Space $U$ | ACC (↑) | Std (↓) |
|---|---|---|---|---|
| w/o $V$ | ✗ | ✓ | 63.3 | 3.06 |
| w/o $U$ | ✓ | ✗ | 75.9 | 3.57 |
| DOP | ✓ | ✓ | 78.2 | 1.63 |

**Linear Module Analysis**: In the main text, our model merges all linear modules using the proposed DOP scheme. In this section, we validate the effectiveness of applying DOP to only a subset of the linear layers. We primarily compare against the state-of-the-art CMM method, OPCM [74]. To this end, we conducted the following ablation experiments: (i) None: Neither OPCM nor DOP is applied to any linear layer. (ii) Q/K/V Projection: OPCM and DOP are applied only to the Q/K/V projection layers. (iii) FFN: OPCM and DOP are applied only to the FFN layers. (iv) ALL: OPCM and DOP are applied to all linear layers. As shown in Tab. 6, we observe that DOP consistently outperforms OPCM. Notably, on the Q/K/V Projection

Table 6: Performance impact analysis of merging different linear layers

| Methods | OPCM | DOP (Ours) |
|---|---|---|
| (i) None | 66.3±0.0 | 66.3±0.0 |
| (ii) Q/K/V Projection | 66.6±0.1 | 72.9±1.8 |
| (iii) FFN | 75.2±0.4 | 75.8±2.0 |
| (iv) ALL | 75.5±0.5 | 78.3±1.6 |

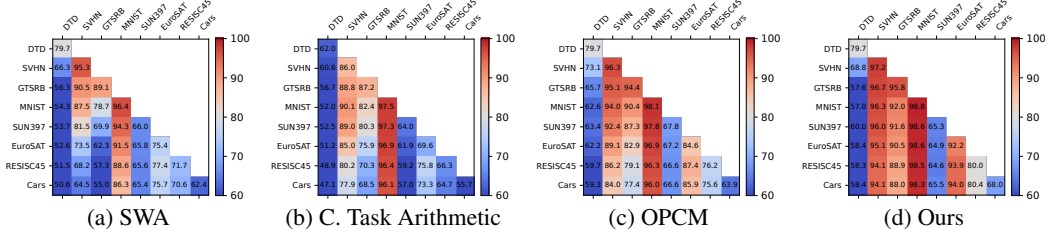

Figure 6: Performance changes during the merging of 8 tasks with the ViT-B/32 architecture. Row $t$ shows test accuracy on all previous tasks $(1-t)$ after merging the $t$-th model (stability), while diagonal values indicate the model's plasticity, reflecting its performance on the newly merged task.

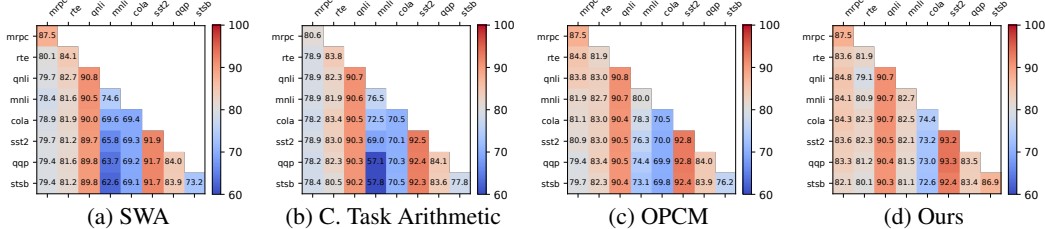

Figure 7: Performance changes during the merging of 8 tasks with the Flan-T5-base architecture. Row $t$ shows test accuracy on all previous tasks $(1-t)$ after merging the $t$-th model (stability), while diagonal values indicate the model's plasticity, reflecting its performance on the newly merged task.

layers, DOP achieves an accuracy of 72.9, compared to 66.6 for OPCM. Furthermore, when optimizing all linear layers, DOP achieves an accuracy of 78.3, surpassing OPCM's 75.5. These results demonstrate that DOP consistently outperforms OPCM across different linear layers.

**Nonlinear Module Analysis**: In the main text, we adopt simple weight averaging [85] as the default strategy for handling nonlinear layers. In this part, we further explored additional strategies for merging nonlinear layers, such as Task Arithmetic [35]. As shown in the Tab. 7, we observe that the performance differences between these two approaches are minor, and both significantly outperform baselines such as SWA and OPCM. This is mainly because the parameters of nonlinear layers constitute only a small proportion of the entire model.

Table 7: Analysis of strategies for merging nonlinear layers

| Methods | ACC |
|---|---|
| Averaging (SWA) | 66.3±0.0 |
| OPCM | 75.5±0.5 |
| Our DOP (Weight Averaging) | 78.3±1.6 |
| Our DOP (Task Arithmetic) | 78.9±1.3 |

**Convergence Analysis**: As shown in Fig. 11, we visualize the optimization trajectory of the stability loss (blue line) and plasticity loss (red line) in the DOP objective across different model merging stages (e.g., stage 8 denotes merging the 8-th incoming model with the previously merged 7 models) and various linear modules (Q/K/V/Out_proj layers in the attention module and FC1/FC2 layers in the feed-forward module). Consistently, both losses exhibit a steady downward trend across all layers and stages, ultimately converging smoothly. This highlights the robust optimization stability of the DOP method.

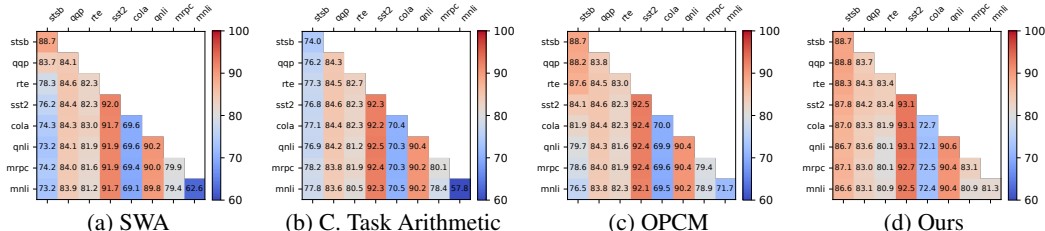

Figure 8: Performance changes during the merging of 8 tasks with the Flan-T5-base architecture. Row $t$ shows test accuracy on all previous tasks $(1-t)$ after merging the $t$-th model (stability), while diagonal values indicate the model's plasticity, reflecting its performance on the newly merged task.

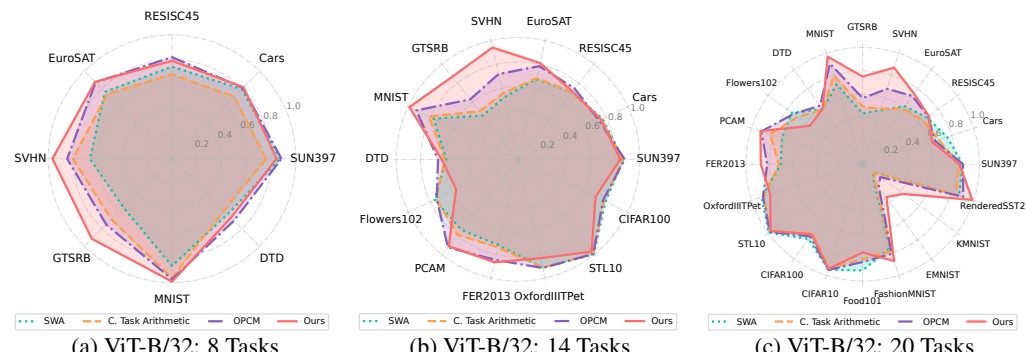

Figure 9: Performance of the final model on all tasks when continually merging different numbers of tasks (8/14/20) using the ViT-B/32 architecture. Each task's accuracy is normalized by its expert model. All results are averaged over ten randomly shuffled task orders.

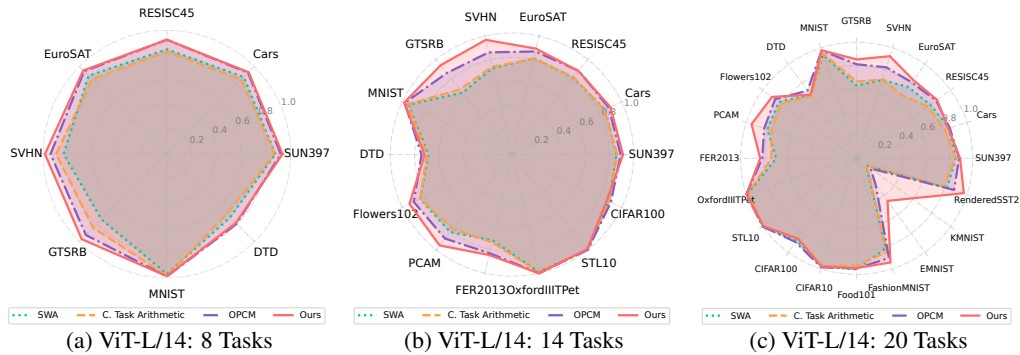

Figure 10: Performance of the final model on all tasks when continually merging different numbers of tasks (8/14/20) using the ViT-L/14 architecture. Each task's accuracy is normalized by its expert model. All results are averaged over ten randomly shuffled task orders.

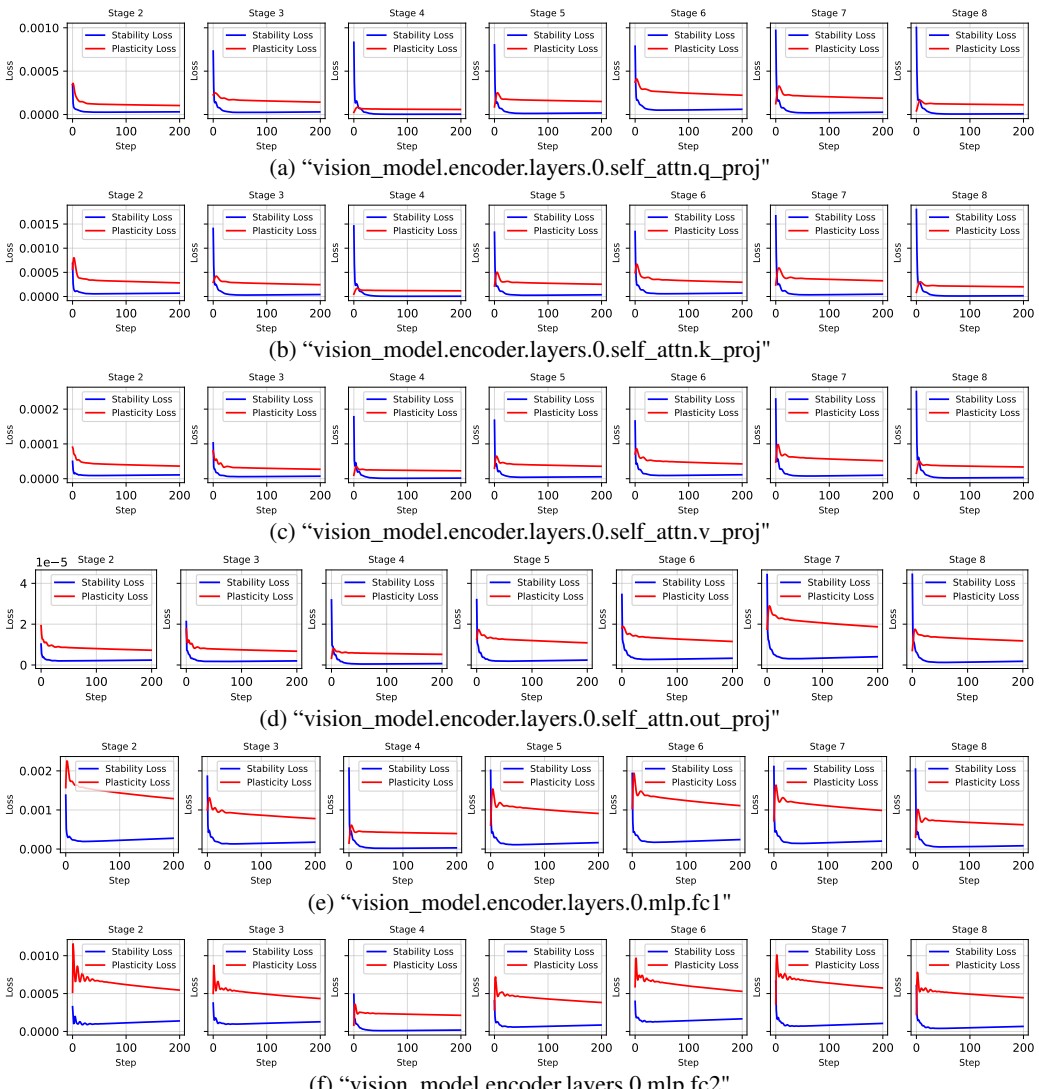

Figure 11: Visualization of the stability and plasticity losses during the optimization process across different linear modules (as shown in Fig. 4(a)) within the first Transformer block of the ViT-B/32 architecture. (a) Q in the Multi-Head Attention module, (b) K in the Multi-Head Attention module, (c) V in the Multi-Head Attention module, (d) Out_Proj in the Multi-Head Attention module, (e) FC1 in the Feed-Forward module, (f) FC2 in the Feed-Forward module. Here, Stage $t$ denotes merging the old model with the $t$-th new task model. We observe that both stability and plasticity losses are low (check the Y-axis scale for reference).

