# OpenReview forum: "Continual Model Merging without Data: Dual Projections for Balancing Stability and Plasticity"
_NeurIPS.cc/2025/Conference — NeurIPS 2025 poster_

### Official Review · Reviewer_fLc1 · 2025-06-04

**Clarity:** 3
**Significance:** 2
**Originality:** 2
**Rating:** 3
**Confidence:** 4

**Summary:**

This paper addresses the problem of continuous model merging, where expert models are received sequentially, and the original training data is not accessible. The authors propose a data-free method named Dual Orthogonal Projection (DOP), which aims to balance stability and plasticity by formulating the merging process as a multi-objective optimization problem. Extensive experiments on both vision and language models across multiple task configurations demonstrate superior performance over baselines.

**Questions:**

See weakness.

**Ethical Concerns:**

["NO or VERY MINOR ethics concerns only"]

**Final Justification:**

After the rebuttal, some of my concerns were addressed. However, I still think this paper makes less contribution compared with existing methods such as OPCM. Moreover, careful discussion with the authors reveals that CMM may not be a significant problem for the model merging community. Therefore, I would like to give the final justification of Weak Reject.

**Limitations:**

No. The author leverages task vectors instead of actual data or gradients to conduct projection. The approximation error remains unsolved and analyzed.

**Paper Formatting Concerns:**

No obvious formatting issue.

**Quality:**

3

**Strengths And Weaknesses:**

Strengths.
+ This paper is well-written and easy to follow.
+ Extensive experiments on various architectures and tasks show the effectiveness of the proposed method.
+ The proposed method does not rely on data during continual merging.

Weaknesses.
+ Although the method is grounded in orthogonal projection theory, this author leverages task vectors instead of actual data or gradients. The approximation error remains unsolved and analyzed. This could affect the reliability of the proposed method.
+ The approach assumes linear layers dominate the parameter space, which is valid for transformers. I suggest the author consider significant non-linear modules, such as bias parameters and parameters of normalization layers.
+ The paper proposes to solve the continuous model merging problem, where old task vectors cannot be stored. However, in practice, storing task vectors is usually not expensive, making this assumption somewhat questionable. So, I think this problem may not be a significant setting for model merging.
+ The novelty is also limited; the proposed method seems very close to OPCM. The proposed method projects boththe  new task vector and the old task vector, while OPCM only project the old task vector.

---

> ### Author Rebuttal · Authors · 2025-07-30
>
> Thank you for your review and affirmation of our work. We'll answer your questions one by one below.
>
> ## Response to Weakness 1: About the "Approximation Error"
> Regarding your concern about our method relying on task vectors instead of actual data or gradients, we would like to provide a detailed explanation as follows:
>
> First, why don't we use real data or true gradients? The continual model merging (CMM) setting [1], like traditional model merging [3,4], assumes that access to the original training data is unavailable. Without access to the data, it is impossible to compute true gradients, so we must approximate them using task vectors to enable effective model merging.
>
> Second, regarding the ability of task vectors to approximate true gradients and the data subspace, we provide a detailed theoretical analysis in Section 3.2.2.
> Specifically, the approximation error of task vectors to the input data in DOP can be theoretically bounded as follows:
>  Let the task vector for a linear layer be $\tau_l$, and the input samples be $\{x_n\}\_{n=1}^{N}$, with each sample undergoing $T$ optimization iterations. Suppose the gradient norm of the loss function with respect to $\tau_l \cdot x_n$ is upper-bounded by $C_l$. Then, there exists a constant $\Phi_l$ such that:
> $
> \| \tau_l - \sum_{n=1}^{N} \alpha_{n,l} (x\_n^\top) \|\_2 \leq \Phi\_l \cdot \left( \sum_{t=1}^{T} \sum_{i=t}^{T} \eta\_t \eta_i \right)
> $,
> where $\eta_t$ denotes the learning rate at iteration $t$, $\alpha_{n,l}$ are reconstruction coefficients, and $\| \cdot \|_2$ is the Euclidean norm. *Due to space limitations, we will provide the detailed proof of this approximation bound in the revised version*.
>
> Furthermore, from the perspective of empirical error, our experiments cover four different complex architectures—including three ViT models of varying sizes for vision tasks. As summarized in the table below, **DOP consistently achieves the best or near-best performance across different architectures and task scales, which strongly validates the effectiveness of using task vectors as gradient approximations in our method**.
>
> In summary, although task vectors are an approximation of the true gradients, in scenarios where training data is inaccessible, they provide a reasonable and efficient solution for model merging.
>
>
> | Method | ViT-B/32 (8 tasks) | ViT-B/32 (14 tasks)  | ViT-B/32 (20 tasks)  | ViT-B/16 (8 tasks) | ViT-B/16 (14 tasks)  | ViT-B/16 (20 tasks)  | ViT-L/14 (8 tasks) | ViT-L/14 (14 tasks)  | ViT-L/14 (20 tasks)  | Flan-T5 (8 tasks)  |
> | :------ | :------: |   :------: |   :------ | :------: |   :------: |   :------: | :------: |   :------: |   :------: |  :------: |
> | SWA  |  66.3 | 65.4 |  61.1 |  72.3   |69.7  | 64.8 |   80.0   |  77.5 |  71.1 |   78.8 |
> | OPCM   |   75.5| 71.9 |  65.7|  81.8   | **77.1** |  70.3|   87.0  | 83.5 | 76.0 |  80.6 |
> | DOP (Ours)   |  **78.3** | **73.0** | **68.5** |  **84.5**   |  76.2 |  **72.2** |  **88.3**   | **85.5** | **80.7** | **83.6**   |
>
>
> ## Response to Weakness 2: About the "Nonlinear Modules"
>
> We agree with your concerns and have supplemented our analysis and experiments on both linear and nonlinear layers to address them.
>
> First, while modern neural network architectures do contain certain nonlinear components, in mainstream architectures such as Transformers, linear layers (e.g., self-attention and feed-forward networks) overwhelmingly dominate the parameter space. As shown in Table 1, the proportion of linear layer parameters exceeds 90% in Flan-T5-Base and all three ViT architectures, with ViT-B/16 reaching as high as 99%. This indicates that, although nonlinear modules exist, **linear layers constitute the vast majority of parameters in these architectures**.
>
> Furthermore, to directly address your concerns about the impact of nonlinear layers (such as bias parameters and parameters of normalization layers), we conducted experiments on various nonlinear layer merging strategies and evaluated their effects on the final model performance. Specifically, we explored several common approaches, including directly using the nonlinear weights from the pretrained model, applying weight averaging, and employing task arithmetic to merge the nonlinear weights of expert models. As shown in Table 2, the performance differences among these three strategies are minimal (79.2, 78.3, and 78.9, respectively). This demonstrates that, in mainstream real-world architectures, **the impact of nonlinear layers on the final merged model performance is limited**, primarily because nonlinear layers account for only a small fraction of the total parameters, while the overall model performance is mainly determined by the linear layers.
>
> In summary, although nonlinear modules play an important role in modern neural networks, the dominance of linear layers in the continual model merging setting means that the merging of nonlinear modules has a relatively limited effect on overall performance.
>
> **Table 1**
> | Architecture | The Proportion of Linear Layer Parameters |
> | :------ | :------: |
> | ViT-B/32  | 97.21% |
> | ViT-B/16  | 99.09% |
> | ViT-L/14  | 99.68% |
> | Flan-T5-Base   | 90.01% |
>
>
> **Table 2**
> | Method | ACC |
> | :------ | :------: |
> | SWA | 66.3 $\pm$ 0.0 |
> | OPCM | 75.5 $\pm$ 0.5 |
> |  |  |
> | Our DOP (Pretrained for NonLinear)  | 79.2 $\pm$ 1.2 |
> | Our DOP (Weight Averaging for NonLinear)  | 78.3 $\pm$ 1.6 |
> | Our DOP (Task Arithmetic for NonLinear)  | 78.9 $\pm$ 1.3 |
>
> ## Response to Weakness 3: About the "Continual Model Merging Setup"
>
> Regarding your concern about the storage overhead of task vectors, we acknowledge that in certain cases (e.g., when model size is small or the number of tasks is limited), the cost of storing task vectors may not be significant. However, the goal of this work is to provide a continual model merging (CMM) solution for memory-constrained environments where access to data is unavailable. **This CMM setting is not introduced for the first time in our paper**; prior studies have also explored similar scenarios [1,2].
>
> Furthermore, traditional model merging aims to combine multiple expert models into a single unified model, thereby reducing storage overhead and enabling knowledge transfer across tasks—a trend that has gained significant traction in recent research [3,4,5,6]. Compared to conventional model merging, the CMM Settings in this work place greater emphasis on scenarios where the model is gradually available in chronological order. Specifically, we introduce additional constraints to make this setting more realistic:
> - (i) Traditional model merging assumes all models are available simultaneously, whereas in practical scenarios, models may become available sequentially. Thus, CMM is more aligned with real-world applications.
> - (ii) Traditional model merging requires loading all models into memory, while CMM only needs to load the pretrained model, the new model, and the merged model at each step, further reducing resource requirements during merging and enhancing practicality.
>
> Overall, we believe that **avoiding the need to store all historical task vectors will greatly improve the scalability and efficiency of the model, especially as the number of tasks or model parameters increases substantially**. In addition, **given the broad applicability of model merging, further optimizing its use in continual scenarios is of significant practical value**.
>
>
> ## Response to Weakness 4: About "Novelty of the Method"
>
> Thank you for your attention to the novelty of our work. At first glance, our method shares some similarities with OPCM, as both leverage projection operations of task vectors to address the continual model merging (CMM) problem. However, the contribution of our proposed DOP method goes far beyond simply extending OPCM's unidirectional projection to a bidirectional one. Specifically, our approach offers **several significant advantages**:
> - **Theoretical Analysis**: DOP provides a deeper theoretical framework by analyzing the relationships among data, gradients, and task vectors, and elucidates their interactions in CMM. Through this analysis, DOP explicitly defines two key objectives—stability and plasticity—from the perspective of orthogonal projection. In contrast, OPCM does not formally define these objectives nor offer a systematic theoretical analysis.
> - **Stability-Plasticity Trade-off**: DOP simultaneously optimizes both stability and plasticity by formulating the problem as a multi-objective optimization, thus achieving a better balance between adapting to new tasks (plasticity) and retaining knowledge from previous tasks (stability). OPCM, on the other hand, focuses solely on stability—aiming to minimize interference from new tasks to old knowledge—while overlooking the crucial aspect of plasticity. Notably, OPCM merely performs a matrix projection operation without truly balancing stability and plasticity through an explicit optimization objective, whereas our method defines a loss function to jointly optimize both goals.
> - **Empirical Performance**: Experimental results across multiple vision and language tasks, four model architectures, and three task scales consistently demonstrate that DOP significantly outperforms OPCM, achieving higher average performance, less forgetting, and stronger task adaptability.
>
> In summary, **the DOP method surpasses OPCM in terms of theoretical depth, methodological soundness, and empirical performance**.
>
> # **Reference**
>
> [1] Merging Models on the Fly Without Retraining: A Sequential Approach to Scalable Continual Model Merging. Arxiv, 2025.
>
> [2] How to Merge Your Multimodal Models Over Time?. CVPR, 2025.
>
> [3] Editing Models with Task Arithmetic. ICLR, 2023.
>
> [4] TIES-Merging: Resolving Interference When Merging Models. NeurIPS, 2023.
>
> [5] Evolutionary optimization of model merging recipes. Nature Machine Intelligence, 2025.
>
> [6] Learning from models beyond fine-tuning. Nature Machine Intelligence, 2025.

---

> ### Comment · Reviewer_fLc1 · 2025-08-01
>
> Thank you for your detailed clarifications. However, I still find it difficult to fully understand the practical motivation and significance of the Continual Model Merging (CMM) setting proposed in your work.
>
> You mention that CMM is designed for memory-constrained environments where access to data is unavailable. However, this justification remains unconvincing to me for two reasons. First, the merging process itself can often be performed offline, where memory resources are generally not a strict limitation. Even if the final model is to be deployed in a memory-constrained environment, the merging step can still happen separately in a server or cloud infrastructure with sufficient resources. Second, while I acknowledge there are cases where task data might not be stored or shared, it seems entirely feasible—and in practice inexpensive—to store the task vectors themselves, which are small, model-agnostic, and generally non-sensitive. As a result, the necessity and impact of avoiding historical task vectors, which is central to your method, still feels insufficiently justified. I believe this is a critical issue: it fundamentally affects whether the proposed work is realistic and valuable, and thus whether it supports a higher score or should lead to a lower one.
>
> Regarding novelty (Weakness 4), I appreciate the effort to clarify the technical contributions beyond OPCM. However, your reply still appears to focus more on the motivation and experimental improvements rather than on concrete algorithmic or theoretical innovation. The main technical difference described is still seems incremental rather than substantially novel.

---

> > ### Author Response · Authors · 2025-08-01
> > **Further clarification on the CMM setting and technical contributions**
> >
> > Dear Reviewer fLc1,
> >
> > Thank you very much for your careful reading of our response and for your quick feedback!
> >
> > ## Response to "Practical Significance of the Continual Model Merging Setting"
> >
> > After carefully reading your reply, we have a better understanding of your concerns regarding the claim that "memory cost is not a real issue". Indeed, for large organizations with ample computational resources, storing task vectors and performing offline merging is relatively straightforward. However, for resource-constrained users, continual model merging (CMM) offers a lower-cost and more convenient solution, which is of practical significance.
> >
> > Most importantly, there are scenarios where CMM is necessary. For example, on the HuggingFace platform:
> > - Developer $U_1$ uploads a merged model $M_{AB}$, which was locally merged from $M_A$ and $M_B$, but does not (or cannot) upload the original $M_A$ and $M_B$.
> > - Subsequently, developer $U_2$ wants to further merge $M_{AB}$ with their own local model $M_C$ (it has already undergone expensive training) to obtain $M_{ABC}$ and publish it. At this point, $U_2$ does not possess and cannot access $M_A$ or $M_B$, so CMM is a better option.
> > - Similarly, if developer $U_3$ wishes to merge $M_{ABC}$ with their own model $M_D$, they can only use CMM, as they do not have access to the earlier models $M_A$, $M_B$, and $M_C$.
> >
> > Therefore, in such continual model reuse and merging scenarios, CMM is a practical and feasible solution.
> >
> >
> > In addition, regarding the statement that "task vectors are small and model-agnostic," this actually depends on the specific context. According to the definition in Task Arithmetic [1], the task vector $\tau_t$ for task $t$ is defined as the difference between the fine-tuned model parameters ($\theta_t$) and the pretrained model parameters ($\theta_0$), i.e., $\tau_t = \theta_t - \theta_0$. Therefore, the size of the task vector is exactly the same as that of the fine-tuned or pretrained model. If the model itself is large, the task vector will also be large.
> >
> >
> > ## Response to "Technical Innovations of Our Method"
> >
> > Thank you for the additional clarification. We have further compared the core technical differences between our DOP and OPCM [2], which can be summarized in the following three aspects:
> > - **New Projection Objects and Directions**: OPCM projects only the new task increment $\Delta W^{(t)}$ onto the subspace orthogonal to the previously merged increment $\Delta W^{(t-1)}_{\text{merged}}$, thereby reducing interference from the new task to existing knowledge. In contrast, our DOP method performs bidirectional projection: similar to OPCM, it ensures that the new increment does not disrupt prior knowledge, but additionally, it projects the previous merged increment onto the subspace orthogonal to the new task increment. This enables the model to achieve both stability and sufficient plasticity to absorb new information.
> > - **Explicit Multi-Objective Loss**: OPCM does not introduce explicit loss terms and relies solely on a matrix projection. However, DOP designs quantifiable projection losses for both stability and plasticity, and jointly minimizes them during optimization: $
> > \min \alpha_s \mathcal{L}_s + \alpha_p \mathcal{L}_p$. Specifically, the two losses are defined as:
> >
> >
> > $\mathcal{L}\_s= ||U\_o^{\top} \Sigma\_o \Delta W\_o^{*} ||\_F^2$
> >
> > $+||\Delta$ $ W\_o^{*} \Sigma$ $\_o V\_o||\_F^2$,
> >
> > $\mathcal{L}\_p = ||U\_n^{\top}  \Sigma\_n \Delta W\_n^{*}||\_F^2$
> >
> > $+ ||\Delta W\_n^{*} \Sigma\_n V\_n||\_F^2$.
> >
> > - **Multi-Objective Loss Balancing Strategy**: OPCM does not consider balancing between the two objectives. DOP introduces multi-gradient descent and exponential moving average  mechanisms, which dynamically adjust $\alpha_s$ and $\alpha_p$ at each iteration according to the gradient directions, thereby achieving a better balance between stability and plasticity.
> >
> > We will further highlight these details in the revised manuscript to better demonstrate the technical value of our work. In addition to the above technical contributions, our paper also provides deeper theoretical analysis and more significant empirical improvements.
> >
> > We sincerely appreciate your thoughtful feedback and valuable suggestions. We hope these clarifications further address your concerns.
> >
> >
> > [1] Editing Models with Task Arithmetic. ICLR, 2023.
> >
> > [2] Merging Models on the Fly Without Retraining: A Sequential Approach to Scalable Continual Model Merging. Arxiv, 2025.

---

> > > ### Comment · Reviewer_fLc1 · 2025-08-04
> > >
> > > Thank you for your detailed explanation. While I understand the practical considerations you've highlighted, I still find it challenging to see the real value of Continual Model Merging (CMM) in many scenarios. For a user like U2, the standard model merging approach — simply merging MAB and MC — seems to be sufficient without introducing the complexity of CMM.
> > >
> > > As for the HuggingFace platform, once a model is uploaded, there are generally methods for creating mirrors or archives, making the need for CMM less critical. The scenarios you mentioned, such as not having access to user-secret models, seem more aligned with federated learning, which could be an interesting or perhaps more meaningful area of research.
> > >
> > > Until there's more concrete evidence of CMM's necessity in real-world applications or clearer use cases where it’s indispensable, I remain unconvinced about its practical value. For now, I can't justify an increase in score on this basis.

---

> > > > ### Author Response · Authors · 2025-08-04
> > > > **Further Explanation on the Continual Model Merging Setting**
> > > >
> > > > Dear Reviewer fLc1,
> > > >
> > > > Thank you again for your careful reading and further feedback!
> > > >
> > > > First, as you pointed out, for user $U_2$, it is indeed possible to simply apply standard model merging, i.e., directly merge models $M_{AB}$ and $M_C$. However, our experimental results demonstrate that this straightforward weight averaging approach has clear performance limitations. As shown in the table below, across four mainstream architectures (ViT-B/32, ViT-B/16, ViT-L/14, and Flan-T5) and three task scales (8, 14, and 20 tasks), traditional weight averaging consistently underperforms our proposed DOP method, with accuracy drops as large as 12.2% (e.g., ViT-B/16 with 8 tasks) and at least 4.8% (e.g., Flan-T5 with 8 tasks). This significant performance gap indicates that simple averaging is insufficient and, while direct, is not the optimal choice. Methods specifically designed for continual model merging settings (such as our DOP) may offer a much better solution.
> > > >
> > > > | Method | ViT-B/32 (8 tasks) | ViT-B/32 (14 tasks)  | ViT-B/32 (20 tasks)  | ViT-B/16 (8 tasks) | ViT-B/16 (14 tasks)  | ViT-B/16 (20 tasks)  | ViT-L/14 (8 tasks) | ViT-L/14 (14 tasks)  | ViT-L/14 (20 tasks)  | Flan-T5 (8 tasks)  |
> > > > | :------ | :------: |   :------: |   :------ | :------: |   :------: |   :------: | :------: |   :------: |   :------: |  :------: |
> > > > | Weight Averaging  |  66.3 | 65.4 |  61.1 |  72.3   |69.7  | 64.8 |   80.0   |  77.5 |  71.1 |   78.8 |
> > > > | DOP (Ours)   |  **78.3** | **73.0** | **68.5** |  **84.5**   |  **76.2** |  **72.2** |  **88.3**   | **85.5** | **80.7** | **83.6**   |
> > > >
> > > > Secondly, regarding the HuggingFace platform example, we emphasize a common scenario: models uploaded to HuggingFace are often locally merged composite models rather than the original expert models. In fact, this is prevalent on HuggingFace's "huggingface.co/spaces/open-llm-leaderboard/open_llm_leaderboard#/", where a dedicated "Merge" tag is used to indicate whether a model has been merged. In such cases, if other users wish to further optimize and merge an existing composite model (for example, by merging their own private model to achieve a higher leaderboard ranking), continual model merging (CMM) becomes a natural and typical solution.
> > > >
> > > > Beyond this, CMM has additional practical applications, such as continually merging checkpoints along a training trajectory to improve out-of-distribution generalization [1,2] and domain generalization [3]. Prior research has shown that models obtained by continual checkpoint merging tend to converge to flatter regions of the loss landscape compared to the final training checkpoint, resulting in better generalization.
> > > >
> > > > In summary, the practical scenarios for continual model merging include, but are not limited to:
> > > > - **Memory-efficient model merging**: Traditional model merging requires loading all candidate models simultaneously, which is often infeasible in resource-constrained environments [4,5,6,7]. CMM provides a lower-cost and more scalable alternative [8,9].
> > > > - **Continual model reuse**: When users wish to further enhance a composite model published by others (as in the HuggingFace example) by merging their own private models, CMM enables continual capability enhancement without requiring access to the original private models of previous users.
> > > > - **Improved model generalization**: Standard training often leads to convergence in sharp regions of the loss landscape, resulting in poor generalization. Continual merging of checkpoints can guide models toward flatter minima, thereby improving generalization [1,2,3].
> > > >
> > > > Finally, we stress that **the continual model merging setting is not introduced for the first time in this paper; rather, our work advances this important and realistic area further**.
> > > >
> > > > We sincerely appreciate your insightful questions and constructive suggestions! In the revised version, we will further highlight the practical significance and potential of continual model merging to make the paper more complete.
> > > >
> > > > **References**:
> > > >
> > > > [1] Stochastic Weight Averaging in Parallel: Large-Batch Training That Generalizes Well. ICLR, 2020.
> > > >
> > > > [2] Averaging weights leads to wider optima and better generalization. UAI, 2018.
> > > >
> > > > [3] Swad: Domain generalization by seeking flat minima. NeurIPS, 2021.
> > > >
> > > > [4] Editing Models with Task Arithmetic. ICLR, 2023.
> > > >
> > > > [5] TIES-Merging: Resolving Interference When Merging Models. NeurIPS, 2023.
> > > >
> > > > [6] Evolutionary optimization of model merging recipes. Nature Machine Intelligence, 2025.
> > > >
> > > > [7] Learning from models beyond fine-tuning. Nature Machine Intelligence, 2025.
> > > >
> > > > [8] How to Merge Your Multimodal Models Over Time?. CVPR, 2025.
> > > >
> > > > [9] Merging Models on the Fly Without Retraining: A Sequential Approach to Scalable Continual Model Merging. Arxiv, 2025.
> > > >
> > > >
> > > > Best regards,
> > > >
> > > > Authors of Submission 7851

---

> > > > > ### Author Response · Authors · 2025-08-07
> > > > > **Continual Merging Setting Clarification**
> > > > >
> > > > > Dear Reviewer fLc1,
> > > > >
> > > > > We sincerely apologize for disturbing you again. As the reviewer-author discussion phase is drawing to a close, we are concerned that you might have missed our further explanation regarding the continual merging setting you cared about amid your busy schedule and numerous email reminders.
> > > > >
> > > > > Could we kindly ask if our latest response has addressed your concerns?
> > > > >
> > > > > Best regards,
> > > > >
> > > > > Authors of Submission 7851

---

> > > > > > ### Comment · Reviewer_fLc1 · 2025-08-09
> > > > > >
> > > > > > Thanks for the reply. My main concern remains whether CMM has practical or potential usefulness. Unfortunately, the current explanation still does not convince me.
> > > > > >
> > > > > > On the first point: As I mentioned initially, for offline model merging, memory is generally not a real bottleneck. Therefore, this motivation is not sufficient to justify the necessity of CMM.
> > > > > >
> > > > > > On the second point: This appears to be more of a hypothetical scenario. Even if it holds, it essentially amounts to a “two-step incremental learning” case, for which a custom two-step merging strategy could be designed, without the need to propose CMM as a general-purpose method.
> > > > > >
> > > > > > On the third point: Improving model generalization is not unique to CMM. Many existing model merging approaches have also demonstrated the ability to enhance generalization performance.
> > > > > >
> > > > > > In addition, I suggest the authors focus on addressing the core question: Does CMM have meaningful research value? Currently, the first CMM paper exists only as a non–peer-reviewed arXiv preprint, and I remain unconvinced that this is a topic worthy of further study.
> > > > > > Moreover, the experimental comparison against only Weight Averaging is insufficient; at minimum, results from representative model merging methods from recent years should be included as a reference. (I am not requesting that the authors provide additional results in this response — rather, I would like to see a clear positioning of the meaningful of CMM.)

---

> > > > > > > ### Author Response · Authors · 2025-08-09
> > > > > > > **Further Clarification on the CMM Setting**
> > > > > > >
> > > > > > > Dear Reviewer fLc1,
> > > > > > >
> > > > > > > Thank you for your further feedback.
> > > > > > >
> > > > > > > Regarding the first point, we believe that the view "memory is generally not a real bottleneck" overlooks some critical constraints in real-world scenarios. While large organizations may have abundant resources, the machine learning community also includes many researchers with limited computational budgets and deployment environments with restricted device memory. In fact, memory-efficient learning has always been an important research direction in the community.
> > > > > > >
> > > > > > > For the second point, the HuggingFace scenario is already a very practical application. As mentioned in our previous comments, many models on HuggingFace are merged models (the open_llm_leaderboard even has a dedicated tag to indicate whether a model is merged), and the original components are often not shared. The need to further merge private new models based on these merged models to continually enhance their capabilities is naturally a continual model merging (CMM) setting.
> > > > > > >
> > > > > > > For the third point, we acknowledge that improving model generalization is not exclusive to CMM. Indeed, we have not claimed that only CMM can improve generalization; we simply cited it as a practical application scenario for continual model merging.
> > > > > > >
> > > > > > > For the fourth point, our comparison with simple "weight averaging" was in response to your comment about "simply merging $M_{AB}$ and $M_C$." In our full experiments, we also compared with several representative model merging methods from recent years (e.g., Task Arithmetic, Ties-Merging, MagMax, OPCM, etc.), and our method consistently achieved the best results.
> > > > > > >
> > > > > > > On the most important question: Does CMM have meaningful research value? Our answer is yes. Compared to traditional model merging, the CMM setting presents many unique research challenges that require further exploration now and in the future:
> > > > > > > - **Stability-plasticity trade-off**: Unlike traditional merging, where all models can be jointly optimized, as analyzed in Section 3.2, CMM must balance the retention of existing knowledge (stability) and adaptation to new knowledge (plasticity) at each sequential step. In other words, if traditional model merging is analogous to multi-task learning, continual model merging is analogous to continual learning.
> > > > > > > - **CMM does not satisfy the commutative property**: Unlike traditional model merging, which is order-independent, in CMM the order in which tasks arrive affects the final performance. This challenge is formally defined as:
> > > > > > > $$
> > > > > > > \mathrm{CMM}\left(\mathrm{CMM}\left(\theta_{\text {merged }}^{(i-1)} ; \theta^{(0)}, \theta^{(i)}\right) ; \theta^{(0)}, \theta^{(i+1)}\right) \neq \mathrm{CMM}\left(\mathrm{CMM}\left(\theta_{\text {merged }}^{(i-1)} ; \theta^{(0)}, \theta^{(i+1)}\right) ; \theta^{(0)}, \theta^{(i)}\right),
> > > > > > > $$
> > > > > > > where $\mathrm{CMM}(\cdot)$ denotes the continual merging algorithm. To avoid this, all our performance comparisons are based on ten random task orders. Developing an order-independent CMM method is a meaningful direction for future work.
> > > > > > > - **Data unavailability**: CMM also inherits the challenge from traditional model merging that original training data is unavailable.
> > > > > > >
> > > > > > > In short, we firmly believe that continual model merging is a topic worthy of further study, and we expect more practical application scenarios to be discovered in the future.
> > > > > > >
> > > > > > > We also understand that there are multiple perspectives on any topic. Finally, we sincerely thank you for all your valuable suggestions and your timely responses throughout the review process!
> > > > > > >
> > > > > > > Thanks!
> > > > > > >
> > > > > > > Best regards,
> > > > > > >
> > > > > > > Authors of Submission 7851

---

> > > > > > > > ### Comment · Reviewer_fLc1 · 2025-08-09
> > > > > > > >
> > > > > > > > Thank you for the response.
> > > > > > > >
> > > > > > > > For the first point. When discussing resource constraints, it is typically computational power, not memory, that is the main issue.  Memory is very very cheap nowadays. As mentioned previously, memory is not typically a bottleneck.
> > > > > > > >
> > > > > > > > For the second point. I am not opposing HuggingFace, but rather pointing out that HuggingFace provides a comprehensive record, and there are numerous mirrored websites storing all checkpoints. This makes Continual Model Merging (CMM) less meaningful, as researchers can simply access every fine-tuned model instead of continually merging them.

---

> > > > > > > > > ### Author Response · Authors · 2025-08-09
> > > > > > > > >
> > > > > > > > > Dear Reviewer fLc1,
> > > > > > > > >
> > > > > > > > > For the first point: “Memory is cheap” does not mean “memory is free”. As the number of models or model scale increases, the memory required for merging will continue to grow.
> > > > > > > > >
> > > > > > > > > For the second point: "The availability of models" does not equate to "the efficient utilization of models", and furthermore, it cannot be guaranteed that all expert models are always accessible simultaneously.
> > > > > > > > >
> > > > > > > > > The value of CMM precisely lies in "how to integrate models in a cost-effective and efficient manner".
> > > > > > > > >
> > > > > > > > > Thanks!
> > > > > > > > >
> > > > > > > > > Best regards,
> > > > > > > > >
> > > > > > > > > Authors of Submission 7851

---

> > > > > > > > > ### Comment · Reviewer_fLc1 · 2025-08-09
> > > > > > > > >
> > > > > > > > > Regarding the first point, as long as you have sufficient computational resources to evaluate the model after merging, the memory cost is trivial compared to the computational cost of evaluation. If you can afford the evaluation computations, the memory required for merging is not a bottleneck and can easily be accommodated.
> > > > > > > > >
> > > > > > > > > As for the second point, you are conflating different concepts. Efficient merging is not exclusive to CMM. Methods such as TiesMerging and CATmerging are also highly efficient and do not require retraining. Therefore, the real value of CMM does not lie in the efficiency of merging, which is already achievable through other methods.

---

> > > > > > > > > > ### Author Response · Authors · 2025-08-09
> > > > > > > > > >
> > > > > > > > > > Dear Reviewer fLc1,
> > > > > > > > > >
> > > > > > > > > > For the first point: If the current machine does not have sufficient memory to store multiple models, is purchasing additional storage devices before starting the work the optimal solution, or should we also consider improving the algorithm?
> > > > > > > > > >
> > > > > > > > > > For the second point: Efficiency refers to the process of merging a new model—traditional methods require loading all models and resolving conflicts among them, while continual model merging (CMM) allows directly loading the previously merged model without re-merging all components.
> > > > > > > > > >
> > > > > > > > > > Best regards,
> > > > > > > > > >
> > > > > > > > > > Authors of Submission 7851

---

### Official Review · Reviewer_4cHS · 2025-06-28

**Clarity:** 3
**Significance:** 3
**Originality:** 3
**Rating:** 4
**Confidence:** 5

**Summary:**

This paper tackles the problem of continual model merging in settings where models are received sequentially, without simultaneous access to all models. It considers the balance between stability and plasticity in continual model merging and investigates the relationships between data, gradients, and task vectors (accumulated gradients). Thus, it proposes a data-free dual orthogonal projection method, using a multi-gradient descent algorithm to derive Pareto-optimal coefficients for stability and plasticity. Experiments validate the effectiveness of the proposed DOP across various models and visual/language tasks.

**Questions:**

1. The paper consider reducing the influence of old merged model on the new model, which means $\Delta W_{n}^{*}X_{n}=0$.

But if the old merged model have similar knowledge or positive knowledge which can be transferred to the new model and make the performance of new model have better performance on the new task data, then $\Delta W_{n}^{*} X_{n}$ may not be zero in this case. How does DOP handle such situations? Does the method risk discarding positively aligned knowledge that would otherwise enhance performance on the new task?

2. The baselines in the experiments are limited. Can authors compare DOP with MagMax[1]?

[1] “MagMax: Leveraging Model Merging for Seamless Continual Learning”, ECCV2024.

3. In Eq(2), why does DOP use $U_{o}^{\top}\Sigma_{o}$ and $\Sigma_{o}V_{o}$ rather than $U_{o}\sqrt{\Sigma_{o}}$ and $\sqrt{\Sigma_{o}}V_{o}^{\top}$, since $U_{o}\sqrt{\Sigma_{o}}\sqrt{\Sigma_{o}}V_{o}^{\top}=\tau_{o}$? Additionally, can authors report the effect of removing singular values (i.e., omitting $\Sigma_o$ and $\Sigma_n$) in this representation? A comparison of these variants could clarify the role of singular value scaling in the projection spaces.

4. What’s the value of final $\alpha_{s,K}$ and $\alpha_{p,K}$? Can authors show the trending or changes of $\alpha_{s,K}$ and $\alpha_{p,K}$ during multi-objective optimization?

5. For computational efficiency, beyond running time, can authors provide additional metrics, such as GPU memory usage or computational complexity, especially in comparison with OPCM?

6. In Eq(3), there appears to be a missing index $k$ in the term  $\nabla_{W_{*,k}}\mathcal{L}_{p}$ in $\min_{\alpha_{k}\in[0,1]}\|\cdot\|$.

**Ethical Concerns:**

["NO or VERY MINOR ethics concerns only"]

**Final Justification:**

I believe the continual merging setting explored in this paper offers a meaningful and under-explored direction that extends beyond traditional assumptions in model merging literature. I keep my positive score.

**Limitations:**

While DOP is a promising algorithm, it is currently limited to full model merging and may not scale efficiently to large language models in resource-constrained environments. Besides, the method is designed to minimize interference from the old merged model, but may ignore scenarios where positive transfer from the old merged model could benefit the new task. And DOP requires training during merging, thus developing a training-free algorithm remains an open direction for future research.

**Quality:**

2

**Strengths And Weaknesses:**

Strengths:

1. This paper proposes DOP, which not only addresses the forgetting issue in continual model merging but also reduces the influence of the old merged model on the new upcoming model.

2. The ablation studies are well-designed and provide clear evidence of the contributions of different components within DOP.

3. The motivation and formulation of DOP are presented clearly, with a well-grounded theoretical explanation.

Weaknesses:

1. The baselines in the experiments are insufficient. For the simultaneous model merging, only three methods are included (SWA, Task Arithmetic, TIES), but there are other recent merging algorithms, such as MagMax[1], which should be considered to strengthen the empirical evaluation.

2. While the paper focuses on proposing an effective algorithm for continual model merging and the experiments also show that the simultaneous model merging algorithms exactly fail in the continual merging setting, it lacks a deeper analysis of why these simultaneous model merging algorithms fail in continual merging. Providing such analysis would reinforce the novelty and necessity of the proposed approach.

3. The proposed DOP requires post-training (gradient-based optimization) when merging, which incurs additional training cost. It would be good to have a training-free merging algorithm instead of additional training.

[1] “MagMax: Leveraging Model Merging for Seamless Continual Learning”, ECCV2024.

---

> ### Author Rebuttal · Authors · 2025-07-30
>
> Thank you very much for your review and affirmation of our work. We'll answer your questions one by one below.
>
> ## Response to Weakness 1 & Question 2: About the "More Baselines"
> Thank you for your suggestion. We have added MagMax[1] as a new baseline and evaluated it across three architectures (ViT-B/32, ViT-B/16, and ViT-L/14) and three task scales (8, 14, and 20 tasks). As shown in the table below, we observe that MagMax outperforms the three baselines—SWA, C. Task Arithmetic, and C. Ties-Merging—but still lags behind OPCM, which is specifically designed for continual model merging, and our proposed DOP. Overall, **our DOP consistently achieves the best performance**.
>
> | Method | ViT-B/32 (8 tasks) | ViT-B/32 (14 tasks)  | ViT-B/32 (20 tasks)  | ViT-B/16 (8 tasks) | ViT-B/16 (14 tasks)  | ViT-B/16 (20 tasks)  | ViT-L/14 (8 tasks) | ViT-L/14 (14 tasks)  | ViT-L/14 (20 tasks)  |
> | :------ | :------: |   :------: |   :------ | :------: |   :------: |   :------: | :------: |   :------: |   :------: |
> | Averaging (SWA)  |  66.3 | 65.4 |  61.1 |  72.3   |69.7  | 64.8 |   80.0   |  77.5 |  71.1 |
> | C. Task Arithmetic   |   67.5 | 66.5 |    60.6 |  77.1 |70.9  | 64.2    |  82.1 |   77.9|  70.3  |
> | C. Ties-Merging    |   49.0 | 66.2 | 59.9 | 66.8    |  70.5| 63.0 | 64.3   | 78.0 | 68.3 |
> | MagMax    | 70.7  | 67.0  | 61.2 |  76.7   | 67.0 | 62.5 |  83.4   |  71.2 | 71.2 |
> | OPCM   |   75.5| 71.9 |  65.7|  81.8   | **77.1** |  70.3|   87.0  | 83.5 | 76.0 |
> | DOP (Ours)   |  **78.3** | **73.0** | **68.5** |  **84.5**   |  76.2 |  **72.2** |  **88.3**   | **85.5** | **80.7** |
>
> [1] “MagMax: Leveraging Model Merging for Seamless Continual Learning”, ECCV 2024.
>
>
> ## Response to Weakness 2: About the "Why Traditional Model Merging Methods Fail"
> Thank you very much for your insightful question. We explain why traditional model merging methods fail in the continual model merging (CMM) setting from two perspectives:
>
> First, traditional methods (such as Ties-Merging) assume simultaneous access to all models and rely on consistent parameter sign alignment across models. However, these approaches face significant limitations in the CMM setting, where tasks arrive sequentially and, at each merging step, only the current model and the previously merged model are accessible. As a result, methods like Ties-Merging **cannot effectively handle scenarios with only local information available at each step**, leading to inaccurate sign alignment and degraded merged model quality.
>
> Another critical challenge is that **continual model merging does not satisfy the commutative property** [1]:
> $$
> \mathrm{CMM}\left(\mathrm{CMM}\left(\theta_{\text {merged }}^{(i-1)} ; \theta^{(0)}, \theta^{(i)}\right) ; \theta^{(0)}, \theta^{(i+1)}\right) \neq \mathrm{CMM}\left(\mathrm{CMM}\left(\theta_{\text {merged }}^{(i-1)} ; \theta^{(0)}, \theta^{(i+1)}\right) ; \theta^{(0)}, \theta^{(i)}\right),
> $$
> where $\mathrm{CMM}(\cdot)$ denotes the continual merging algorithm. This formula highlights that, in continual model merging, the order of merging affects the final result, which explains the performance degradation of traditional model merging methods in this scenario.
>
> [1] "Merging Models on the Fly Without Retraining: A Sequential Approach to Scalable Continual Model Merging", Arxiv, 2025.
>
>
> ## Response to Weakness 3: About the "Training of DOP Algorithm"
> Regarding the concern that DOP requires gradient-based optimization, we would like to clarify the following points. Although DOP does involve gradient computation during optimization, it offers several key advantages that keep the extra training cost low and ensure high computational efficiency:
> - **No Forward Pass Required**: DOP does not rely on the original task data. Instead, it directly optimizes based on the parameter differences between models. This means that, during merging, there is no need to perform traditional forward passes to compute losses or update gradients.
> - **No Backpropagation Through the Whole Network**: DOP performs optimization layer by layer, i.e., it independently optimizes each linear layer rather than jointly optimizing the entire model. This approach eliminates the need for complex backpropagation through the entire network, resulting in significant savings in both time and GPU memory.
>
> In summary, while DOP requires gradient computation, **it is highly efficient, does not depend on original data, and avoids the computational burden of full backpropagation by optimizing each layer independently**. As shown in the table below, the optimization time for merging each linear layer is very short, typically only 1 to 3 seconds.
>
> | Architecture | Execution Time|
> | :------ | :------: |
> | ViT-B/32 |  1.74 $\pm$ 1.18 |
> | ViT-B/16 |  1.55 $\pm$ 0.47 |
> | ViT-L/14 |  3.11 $\pm$ 1.73 |
> | Flan-T5-Base |  1.30 $\pm$ 1.32 |
>
> ## Response to Question 1: About the "Projection in Scenarios Where the Old Model Helps the New Model"
> In our DOP method, although we minimize the parameter differences between the old and new models to reduce interference from previous tasks, we do not completely disregard the beneficial knowledge contained in the old model (in practice, $\Delta W_{n}^{*}X_n$ does not converge perfectly to zero, but only approaches this extreme value). More importantly, when computing gradients and performing optimization, DOP adjusts the parameter differences via orthogonal projection, thereby avoiding excessive interference with the new task. In this process, if some knowledge from the old model aligns with the objectives of the new task, DOP will retain this information, as it resides in a shared subspace and is thus unaffected by the projection.
>
> ## Response to Question 3: About the "Analysis of Variants of Projection Space"
> Thank you very much for your suggestion. We analyzed the performance of the following two variants on the ViT-B/32 architecture:
> - (i) DOP (w/o $\Sigma$) refers to directly removing the singular values $\Sigma$. We observed that its performance drops significantly (although it still outperforms the baseline method SWA), as it discards the projection penalty along the principal directions (those corresponding to larger singular values).
> - (ii) DOP ($\sqrt{\Sigma}$) follows your suggestion of applying a square root to the singular values, which still preserves the relative importance across different directions. As shown in the table below, we observe that the results using $\sqrt{\Sigma}$ are essentially the same as those using $\Sigma$ (78.2% vs. 78.3%).
>
> In summary, **retaining the magnitude of the singular values as the relative penalty strength for each projection direction is important**.
>
> | Method | ACC |
> | :------ | :------: |
> | SWA | 66.3 $\pm$ 0.0 |
> | OPCM | 75.5 $\pm$ 0.5 |
> |  |  |
> | DOP (w/o $\Sigma$)  |  67.6 $\pm$ 1.3 |
> | DOP ($\sqrt{\Sigma}$)  | 78.2 $\pm$ 1.5  |
> | DOP (Ours)  |  78.3 $\pm$ 1.6 |
>
>
> ## Response to Question 4: About the "Variation of Coefficient Alpha"
> In our method, the coefficients $\alpha = \{\alpha_{s,k}, \alpha_{p,k}\}$ are adaptively updated across different layers, parameter modules, and merging stages. To further analyze the dynamic evolution of $\alpha$ during optimization, we visualize the values of $\{\alpha_{s,k}, \alpha_{p,k}\}$ at various training steps.
>
> For illustration, we randomly select two linear modules from two different stages, and record the values of $\alpha$ at different training steps ($k=0,25,\ldots,200$). As shown in the following tables, we consistently observe that the stability coefficient $\alpha_{s,k}$ gradually decreases from the initial value of 0.8, while the plasticity coefficient $\alpha_{p,k}$ increases from 0.2 to around 0.3. This trend reflects that, in the early stages, the model tends to retain more knowledge from the merged model, and as training progresses, it gradually incorporates new knowledge, resulting in a decrease in $\alpha_{s,k}$ and an increase in $\alpha_{p,k}$.
>
>
> Stage 2 - vision_model.encoder.layers.11.self_attn.k_proj:
> | Iteration (k) | 0 | 25 | 50 | 75 | 100 | 125  | 150 | 175  | 200  |
> | :------ | :------: |  :------: |  :------: |  :------: |  :------: |  :------: |  :------: |  :------: |  :------: |
> | $\alpha_{s,k}$ | 0.80  |  0.79  |  0.78  | 0.78  |  0.77 | 0.77  | 0.76  |  0.76  |  0.75 |
> | $\alpha_{p,k}$ | 0.20  |  0.21   | 0.22   |  0.22 |   0.23 | 0.23  | 0.24  |  0.24  |  0.25 |
>
>
> Stage 8 - vision_model.encoder.layers.11.self_attn.v_proj:
> | Iteration (k) | 0 | 25 | 50 | 75 | 100 | 125  | 150 | 175  | 200  |
> | :------ | :------: |  :------: |  :------: |  :------: |  :------: |  :------: |  :------: |  :------: |  :------: |
> | $\alpha_{s,k}$ | 0.80  |  0.78  |  0.77  | 0.75  |  0.74 | 0.73  | 0.72  |  0.71  |  0.70 |
> | $\alpha_{p,k}$ | 0.20  |  0.22   | 0.23   |  0.25 |   0.26 | 0.27  | 0.28  |  0.29  |  0.30 |
>
> ## Response to Question 5: About "More Metrics on Computational Efficiency"
> Thank you for your suggestion. To provide a more comprehensive evaluation, we have included the peak GPU memory usage (Peak GPU Memory Allocated) during both the model merging and inference stages on the ViT-B/32 architecture. As shown in the table below, during the merging stage, our method requires slightly more GPU memory than OPCM due to the optimization performed on each linear layer. However, the overall GPU consumption remains very low, at only 247.3 MB. In the inference stage, DOP and OPCM exhibit comparable GPU memory usage. **While previous results have demonstrated the time efficiency of our method during optimization, these findings further confirm that our approach is also highly efficient in terms of memory usage.**
>
> | Method | Merging Stage (GPU) | Testing Stage (GPU) |
> | :------ | :------: |  :------: |
> | OPCM  |  71.12 MB |  964.17 MB |
> | DOP (Ours)  | 247.3 MB  | 973.48 MB  |
>
> ## Response to Q6: About the "Typo Error in Eq(3)"
> We have carefully reviewed and corrected it in the revised version.

---

> > ### Comment · Reviewer_4cHS · 2025-08-04
> >
> > Thank the authors for taking the effort in the rebuttal. The responses have addressed most of my concerns. Continual modeling merging is a promising paradigm. Model merging is to merge task-specific models to obtain a multi-task model, but almost all existing approaches assume simultaneous access to all models. This assumption does not hold in sequential task stream scenarios where models are obtained over time. [1] has shown that previous merging methods do not generalize well to the temporal merging setting. Continual Model Merging (CMM) specifically addresses this more practical and challenging setting where models are merged sequentially, potentially after all old models are discarded or access is restricted.
> >
> > [1] How to Merge Your Multimodal Models Over Time?, CVPR2025.
> >
> > Therefore, I believe the continual merging setting explored in this paper offers a meaningful and under-explored direction that extends beyond traditional assumptions in model merging literature. I keep my positive score.

---

> > > ### Author Response · Authors · 2025-08-04
> > > **Thank you for your suggestions and your support!**
> > >
> > > Dear Reviewer 4cHS,
> > >
> > > Thank you very much for your dedication to our work and all your valuable suggestions.
> > >
> > > We also appreciate your recognition that "continual model merging is a promising paradigm" and your comment that "continual merging setting explored in this paper offers a meaningful and under-explored direction."
> > >
> > > Thanks!
> > >
> > > Best regards,
> > >
> > > Authors of Submission 7851

---

### Official Review · Reviewer_1H81 · 2025-07-01

**Clarity:** 3
**Significance:** 3
**Originality:** 2
**Rating:** 4
**Confidence:** 4

**Summary:**

This paper proposes a Dual Orthogonal Projection (DOP) method for Continual Model Merging (CMM) without access to original data. DOP formally defines and balances stability and plasticity by projecting parameter updates onto the subspaces of previous and new tasks. The approach is formulated as a multi-objective optimization and solved with multi-gradient descent. Experiments on vision and language models show that DOP outperforms existing methods in both effectiveness and efficiency.

**Questions:**

- For continual model merging, is it possible to maintain a streamlined model library, where each model covers a certain category, and combinations of models can cover more cases? Could a dynamic merging mechanism based on this model library be used to address the formulation problem?
- Can the current orthogonal projection analysis be generalized to more complex network architectures?
- Can the proposed method be extended to networks with different structures?

**Ethical Concerns:**

["NO or VERY MINOR ethics concerns only"]

**Final Justification:**

After reading the author's response, some of my concerns have been solved. I keep my previous justification of borderline acceptance.

**Limitations:**

Yes

**Quality:**

3

**Strengths And Weaknesses:**

Strengths
- Analyzing continual model merging from the perspective of orthogonal projection is interesting.
- The high-level idea of the method is intuitive: the space spanned by model gradient updates is used to approximate the input space, and by ensuring that the parameter changes for new and old tasks are orthogonally projected onto their respective input spaces, the method achieves both retention of previous knowledge and learning of new information.

Weaknesses
- The analysis of orthogonal projection is derived under the assumption of linear functions, but neural networks contain many nonlinear components and are highly stacked. It is unclear how the current analysis can be extended to structures that are closer to real scenarios, such as linear layers with activation functions or multiple stacked linear layers.
- More baseline comparisons could be included in the experimental results.

---

> ### Author Rebuttal · Authors · 2025-07-30
>
> Thank you very much for your review and affirmation of our work. We'll answer your questions one by one below.
>
> ## Response to Weakness 1 \& Question 2: About the "Analysis under Complex Network Architectures"
> We understand your concerns regarding the linearity assumption, especially given that neural networks typically contain nonlinear activation functions and multiple stacked layers. We would like to clarify that, although modern neural network architectures generally include both linear and nonlinear components, mainstream architectures such as Transformers are in fact dominated by a series of linear layers (e.g., embedding layers, self-attention layers, and feed-forward networks). These linear layers constitute the majority of the parameters in the overall architecture. As shown in the table below, the proportion of linear layer parameters exceeds 90% in both Flan-T5 and ViT architectures, reaching as high as 99% in some cases.
>
> Moreover, all experiments in this paper are conducted on real-world, complex network architectures (including ViT-B/32, ViT-B/16, ViT-L/14, and Flan-T5, which are widely used vision and language models). These architectures contain multiple stacked linear layers, nonlinear activation functions, and other nonlinear components. **Our method consistently outperforms baseline methods across all these real architectures, demonstrating its applicability to complex, practical network settings.**
>
> On the other hand, to address your concerns regarding the treatment of nonlinear layers, we have implemented and compared different merging strategies for nonlinear layers to observe their impact on final performance. Specifically, we experimented with several approaches, such as directly using the nonlinear weights from the pretrained model, applying weight averaging, and employing task arithmetic to merge the nonlinear weights of expert models. As shown in the table below, the performance differences among these strategies are minimal (79.2% vs. 78.3% vs. 78.9%), and all consistently outperform baselines such as SWA and OPCM. This indicates that **the impact of nonlinear layers on the merged model's performance is limited, as they account for only a small proportion of the total parameters**.
>
> | Architecture | The Proportion of Linear Layer Parameters |
> | :------ | :------: |
> | ViT-B/32  | 97.21% |
> | ViT-B/16  | 99.09% |
> | ViT-L/14  | 99.68% |
> | Flan-T5-Base   | 90.01% |
>
> | Method | ACC |
> | :------ | :------: |
> | SWA | 66.3 $\pm$ 0.0 |
> | OPCM | 75.5 $\pm$ 0.5 |
> |  | |
> | Our DOP (Pretrained for NonLinear)  | 79.2 $\pm$ 1.2 |
> | Our DOP (Weight Averaging for NonLinear)  | 78.3 $\pm$ 1.6 |
> | Our DOP (Task Arithmetic for NonLinear)  | 78.9 $\pm$ 1.3 |
>
>
>
> ## Response to Weakness 2: About the "More Baselines"
>
> Thank you for your suggestion. We have further included MagMax[1] as a new baseline and evaluated it across three architectures (ViT-B/32, ViT-B/16, and ViT-L/14) and three task scales (8, 14, and 20 tasks). As shown in the table below, we observe that MagMax outperforms the three baselines—SWA, C. Task Arithmetic, and C. Ties-Merging—but still lags behind OPCM, which is specifically designed for continual model merging, and our proposed DOP. Overall, **our DOP consistently achieves the best performance**.
>
> | Method | ViT-B/32 (8 tasks) | ViT-B/32 (14 tasks)  | ViT-B/32 (20 tasks)  | ViT-B/16 (8 tasks) | ViT-B/16 (14 tasks)  | ViT-B/16 (20 tasks)  | ViT-L/14 (8 tasks) | ViT-L/14 (14 tasks)  | ViT-L/14 (20 tasks)  |
> | :------ | :------: |   :------: |   :------ | :------: |   :------: |   :------: | :------: |   :------: |   :------: |
> | Averaging (SWA)  |  66.3 | 65.4 |  61.1 |  72.3   |69.7  | 64.8 |   80.0   |  77.5 |  71.1 |
> | C. Task Arithmetic   |   67.5 | 66.5 |    60.6 |  77.1 |70.9  | 64.2    |  82.1 |   77.9|  70.3  |
> | C. Ties-Merging    |   49.0 | 66.2 | 59.9 | 66.8    |  70.5| 63.0 | 64.3   | 78.0 | 68.3 |
> | MagMax    | 70.7  | 67.0  | 61.2 |  76.7   | 67.0 | 62.5 |  83.4   |  71.2 | 71.2 |
> | OPCM   |   75.5| 71.9 |  65.7|  81.8   | **77.1** |  70.3|   87.0  | 83.5 | 76.0 |
> | DOP (Ours)   |  **78.3** | **73.0** | **68.5** |  **84.5**   |  76.2 |  **72.2** |  **88.3**   | **85.5** | **80.7** |
>
> [1] “MagMax: Leveraging Model Merging for Seamless Continual Learning”, ECCV 2024.
>
>
>
> ## Response to Question 1: About the "Dynamic Model Merging Library"
> We agree with your perspective that maintaining a model library and leveraging model composition to cover more tasks or categories is indeed a promising direction. This approach is closely related to ensemble learning, which aims to collaboratively solve complex tasks by integrating the capabilities of multiple experts. However, the continual model merging (CMM) setting discussed in this paper is more aligned with multi-task learning or continual learning, where only a single set of model parameters is maintained to accomplish all tasks, resulting in higher efficiency. In summary, dynamic model libraries and unified models have different application scenarios:
>
> - **Expert Library or Ensemble Learning**: The expert library approach is suitable for scenarios with ample memory resources, where each model specializes in different tasks or categories, and model composition is used to enhance overall task coverage. The advantage of this method lies in its ability to handle more complex tasks through the collaboration of multiple expert models. However, it requires significant memory to store multiple models and may lead to inefficiency, especially in resource-constrained environments.
> - **Continual Model Merging**: In contrast, continual model merging is more appropriate for resource-limited scenarios, particularly when memory or computational capacity is restricted. In the continual model merging setting, the goal is to build a single unified model that incorporates the capabilities of all tasks by sequentially merging models for new tasks as they arrive. This approach avoids the need to store a large number of expert models, thereby reducing memory consumption while maintaining adaptability to new tasks and retention of previous knowledge.
>
> Therefore, expert libraries and continual model merging serve different application needs. We believe that **each has its own advantages and limitations, and the appropriate strategy can be chosen based on actual resource constraints and task requirements**.
>
>
>
> ## Response to Question 3: About the "Extended to Networks with Different Structures"
> In our experiments, we include four different architectures: three ViT models of varying sizes for vision tasks (ViT-B/32, ViT-B/16, and ViT-L/14) and the Flan-T5-Base model for language tasks. Additionally, our experiments cover different numbers of tasks (8, 14, and 20). As summarized in the table below, **DOP consistently achieves the best or near-best performance across different architectures and task scales, which strongly demonstrates the effectiveness of our method under various complex architectures**.
>
> | Method | ViT-B/32 (8 tasks) | ViT-B/32 (14 tasks)  | ViT-B/32 (20 tasks)  | ViT-B/16 (8 tasks) | ViT-B/16 (14 tasks)  | ViT-B/16 (20 tasks)  | ViT-L/14 (8 tasks) | ViT-L/14 (14 tasks)  | ViT-L/14 (20 tasks)  | Flan-T5-Base (8 tasks)  |
> | :------ | :------: |   :------: |   :------ | :------: |   :------: |   :------: | :------: |   :------: |   :------: |  :------: |
> | SWA  |  66.3 | 65.4 |  61.1 |  72.3   |69.7  | 64.8 |   80.0   |  77.5 |  71.1 |   78.8 |
> | OPCM   |   75.5| 71.9 |  65.7|  81.8   | **77.1** |  70.3|   87.0  | 83.5 | 76.0 |  80.6 |
> | DOP (Ours)   |  **78.3** | **73.0** | **68.5** |  **84.5**   |  76.2 |  **72.2** |  **88.3**   | **85.5** | **80.7** | **83.6**   |

---

> > ### Comment · Reviewer_1H81 · 2025-08-04
> >
> > Thank you for the author's response. I keep my positive score.

---

> > > ### Author Response · Authors · 2025-08-04
> > > **Thank you for your support!**
> > >
> > > Dear Reviewer 1H81,
> > >
> > > Thank you very much for your valuable suggestions and support for our work!
> > >
> > > Best regards,
> > >
> > > Authors of Submission 7851

---

> ### Comment · Reviewer_1H81 · 2025-08-06
>
> For the rebuttal, I have a question that while linear layers constitute the dominant component of most existing neural network architectures, the non-linear activation functions play a crucial role in determining the model's overall capability. Given this, can the current theory be extended to incorporate both the single linear layer and non-linear activation function, such as ReLU?

---

> > ### Author Response · Authors · 2025-08-06
> > **Clarification on nonlinear activation functions**
> >
> > Dear Reviewer 1H81,
> >
> > Certainly, the theoretical foundations of our work remain valid even when nonlinear activation functions are present.
> >
> > Since activation functions themselves do not contain any parameters that require merging, our orthogonal projection definitions and merging procedures are always focused on the parameters of linear layers. Specifically, as long as the merged parameters $w_{*}$ produce the same pre-activation outputs as the original parameters $w_{o}$ (or $w_{n}$) on the input subspace of the old (or new) task via orthogonal projection, the outputs after any element-wise activation (such as GELU, ReLU, or Sigmoid) will also be identical. Moreover, our subspace relationship analysis already accounts for the impact of nonlinear mappings on the inclusion relationships among gradients and task vector subspaces.
> >
> > Finally, **all models used in our experiments include nonlinear activations**; for example, the ViT [1] architecture employs GELU [2] as its activation function. The results demonstrate that the theory and methods proposed in this paper are applicable to networks with activation functions.
> >
> > [1] Dosovitskiy, Alexey, et al. "An image is worth 16x16 words: Transformers for image recognition at scale." ICLR, 2021.
> >
> > [2] Hendrycks, Dan, and Kevin Gimpel. "Gaussian error linear units (gelus)." arXiv preprint arXiv:1606.08415 (2016).
> >
> > Best regards,
> >
> > Authors of Submission 7851

---

> > > ### Comment · Reviewer_1H81 · 2025-08-06
> > >
> > > Thank you for the author's response; my question has been resolved.

---

### Official Review · Reviewer_munj · 2025-07-03

**Clarity:** 4
**Significance:** 4
**Originality:** 4
**Rating:** 4
**Confidence:** 5

**Summary:**

This paper addresses the problem of continual model merging (CMM), where models are merged incrementally without access to original training data. The authors propose Dual Orthogonal Projection (DOP), a data-free method that balances stability (retaining old task knowledge) and plasticity (adapting to new tasks) through multi-objective optimization. Key contributions include formalizing stability and plasticity through orthogonal projection theory, analyzing subspace relationships, and introducing DOP with MGDA optimization.

**Questions:**

Questions
1. OPCM is also a data-free method based on orthogonal projections. Are there additional advantages of DOP beyond the use of MGDA for adaptively balancing the weights between old and new tasks?
2. In the Tab.4 of ablation studies for projection space, it seems that the orthogonal constraints in row space and column space have huge differences (63.3 v.s. 75.9). Could authors explain this phenomenon?
3. Fig.3(d) highlights the importance of the coefficient alpha. Could the authors visualize the solved alpha during optimization and give more insights into the model merging process?
4. Linear components mainly reside in embedding layers, Q/K/V projection layers, and FFN layers. Does DOP demonstrate varying degrees of improvement in these three parts compared to SWA and OPCM?

**Ethical Concerns:**

["NO or VERY MINOR ethics concerns only"]

**Final Justification:**

The authors' response sufficiently answered my questions. I will keep my positive rating.

**Limitations:**

Yes.

**Paper Formatting Concerns:**

None.

**Quality:**

4

**Strengths And Weaknesses:**

Strengths
1. This work proposes a novel perspective of the relationships among the subspaces spanned by task data, gradients, and task vectors.
2. DOP is a data-free method, which is both memory- and time-efficient.
3. Extensive experiments on vision and language tasks demonstrate superior performance over SOTA methods.
Weaknesses
1. Non-linear components are handled via simple averaging, while different linear layers are not differentiated, which may still leave room for further optimization.

---

> ### Author Rebuttal · Authors · 2025-07-30
>
> Thank you very much for your review and affirmation of our work. We'll answer your questions one by one below.
>
> ## Response to Weakness 1:  About "Nonlinear Layer Processing Methods"
> Thank you for your suggestion regarding the treatment of nonlinear layers. In the main text, we have by default adopted simple weight averaging for handling nonlinear layers. Following your advice, we further explored additional strategies for merging nonlinear layers, including Weight Averaging and Task Arithmetic[1]. As shown in the table below, we observe that **the performance differences between these two approaches are minor**, and both significantly outperform baselines such as SWA and OPCM. This is mainly because the parameters of nonlinear layers constitute only a small proportion of the entire model.
>
> | Method | ACC |
> | :------ | :------: |
> | SWA | 66.3 $\pm$ 0.0 |
> | OPCM | 75.5 $\pm$ 0.5 |
> |  | |
> | Our DOP (Weight Averaging for NonLinear)  | 78.3 $\pm$ 1.6 |
> | Our DOP (Task Arithmetic for NonLinear)  | 78.9 $\pm$ 1.3 |
>
> [1] Editing Models with Task Arithmetic. ICLR, 2023.
>
>
> ## Response to Question 1: About "Advantages of our DOP Compared to OPCM"
> Compared to OPCM, the proposed DOP method demonstrates clear advantages in several aspects:
> - **Theoretical Analysis**: DOP provides a more comprehensive theoretical framework by analyzing the relationships among data, gradients, and task vectors in the context of continual model merging (CMM). Through this analysis, DOP explicitly defines two key objectives—stability and plasticity—from the perspective of orthogonal projection. In contrast, OPCM does not formally define these objectives nor provide such theoretical analysis.
> - **Stability-Plasticity Trade-off**: DOP simultaneously optimizes both stability and plasticity by formulating the problem as a multi-objective optimization, thus achieving a better balance between adapting to new tasks (plasticity) and retaining knowledge from previous tasks (stability). OPCM, however, focuses only on stability, aiming to minimize interference from new tasks to old knowledge, and overlooks the important aspect of plasticity.
> - **Empirical Performance**: Experimental results across multiple vision and language tasks, four model architectures, and three task scales consistently show that DOP outperforms OPCM, achieving higher average performance and less forgetting.
>
> In summary, **DOP not only offers a more rigorous theoretical foundation and methodological soundness for continual model merging, but also demonstrates superior empirical performance in practical applications**.
>
>
> ## Response to Question 2: About "Differences in Projection Space Construction"
> Thank you very much for your insightful question. We believe that the row space and column space possess distinct geometric properties in capturing the structure and variations of task data. Specifically, the row space primarily reflects the variation of task vectors in the input data (feature space), while the column space directly reveals the variation of task vectors in the model parameters (output space). As discussed in our main text and supported by previous works on orthogonal projection, traditional orthogonal projection operations in continual learning are typically performed in the input space [1,2], which aligns with our results. Therefore, compared to the output space, projection in the input space may be more effective, as it better captures critical information from the input data and thus impacts the overall model performance. Furthermore, simultaneously capturing the characteristics of both input and output spaces is superior to considering either alone.
>
> [1] Gradient projection memory for continual learning. ICLR, 2021.
>
> [2] TRGP: Trust region gradient projection for continual learning. ICLR, 2022.
>
>
> ## Response to Question 3: About "Visualization of Coefficient Alpha"
> In our method, the coefficients $\alpha = \{\alpha_{s,k}, \alpha_{p,k}\}$ are adaptively updated across different layers, parameter modules, and merging stages. To further analyze the dynamic evolution of $\alpha$ during optimization, we visualize the values of $\{\alpha_{s,k}, \alpha_{p,k}\}$ at various training steps.
>
> For illustration, we randomly select three linear modules from three different stages (Stage 2 - vision_model.encoder.layers.11.self_attn.k_proj, Stage 4 - vision_model.encoder.layers.0.self_attn.k_proj, and Stage 8 - vision_model.encoder.layers.11.self_attn.v_proj), and record the values of $\alpha$ at different training steps ($k=0,25,50,75,100,125,150,175,200$). As shown in the following tables, we consistently observe that the stability coefficient $\alpha_{s,k}$ gradually decreases from the initial value of 0.8, while the plasticity coefficient $\alpha_{p,k}$ increases from 0.2 to around 0.3. This trend reflects that, in the early stages, the model tends to retain more knowledge from the merged model, and as training progresses, it gradually incorporates new knowledge, resulting in a decrease in $\alpha_{s,k}$ and an increase in $\alpha_{p,k}$.
>
> We will provide detailed plots of the evolution of $\alpha = \{\alpha_{s,k}, \alpha_{p,k}\}$ at each iteration step in the revised version.
>
> Stage 2 - vision_model.encoder.layers.11.self_attn.k_proj:
> | Iteration (k) | 0 | 25 | 50 | 75 | 100 | 125  | 150 | 175  | 200  |
> | :------ | :------: |  :------: |  :------: |  :------: |  :------: |  :------: |  :------: |  :------: |  :------: |
> | $\alpha_{s,k}$ | 0.80  |  0.79  |  0.78  | 0.78  |  0.77 | 0.77  | 0.76  |  0.76  |  0.75 |
> | $\alpha_{p,k}$ | 0.20  |  0.21   | 0.22   |  0.22 |   0.23 | 0.23  | 0.24  |  0.24  |  0.25 |
>
> Stage 4 - vision_model.encoder.layers.0.self_attn.k_proj:
> | Iteration (k) | 0 | 25 | 50 | 75 | 100 | 125  | 150 | 175  | 200  |
> | :------ | :------: |  :------: |  :------: |  :------: |  :------: |  :------: |  :------: |  :------: |  :------: |
> | $\alpha_{s,k}$ | 0.80  |  0.78  |  0.77  | 0.75  |  0.73 | 0.72  | 0.70  |  0.69  |  0.67 |
> | $\alpha_{p,k}$ | 0.20  |  0.22   | 0.23   |  0.25 |   0.27 | 0.28  | 0.30  |  0.31  |  0.33 |
>
> Stage 8 - vision_model.encoder.layers.11.self_attn.v_proj:
> | Iteration (k) | 0 | 25 | 50 | 75 | 100 | 125  | 150 | 175  | 200  |
> | :------ | :------: |  :------: |  :------: |  :------: |  :------: |  :------: |  :------: |  :------: |  :------: |
> | $\alpha_{s,k}$ | 0.80  |  0.78  |  0.77  | 0.75  |  0.74 | 0.73  | 0.72  |  0.71  |  0.70 |
> | $\alpha_{p,k}$ | 0.20  |  0.22   | 0.23   |  0.25 |   0.26 | 0.27  | 0.28  |  0.29  |  0.30 |
>
>
> ## Response to Question 4: About "Improvements of DOP in Different Linear Layers"
> Thank you very much for raising this insightful question. We agree that analyzing whether DOP consistently outperforms OPCM across different linear submodules is highly valuable. To this end, we conducted the following ablation experiments:
> - (i) None: Neither OPCM nor DOP is applied to any linear layer.
> - (ii) Q/K/V Projection: OPCM and DOP are applied only to the Q/K/V projection layers.
> - (iii) FFN: OPCM and DOP are applied only to the FFN layers.
> - (iv) ALL: OPCM and DOP are applied to all linear layers.
>
> We observe that DOP consistently outperforms OPCM. Notably, on the Q/K/V Projection layers, DOP achieves an accuracy of 72.9, compared to 66.6 for OPCM. Furthermore, when optimizing all linear layers, DOP achieves an accuracy of 78.3, surpassing OPCM's 75.5. **These results demonstrate that DOP consistently outperforms OPCM across different linear layers.**
>
> | Method | (i) None |   (ii) Q/K/V Projection |  (iii) FFN |  (iv) ALL |
> | :------ |  :------: |  :------: |  :------: |   :------: |
> | OPCM   | 66.3 | 66.6 $\pm$ 0.1 | 75.2 $\pm$ 0.4 | 75.5 $\pm$ 0.5 |
> | DOP (Ours)  | 66.3 | **72.9 $\pm$ 1.8** | **75.8 $\pm$ 2.0** | **78.3 $\pm$ 1.6** |

---

> > ### Comment · Reviewer_munj · 2025-08-06
> >
> > Thanks for the responses. I will keep my positive score.

---

> > > ### Author Response · Authors · 2025-08-06
> > > **Thank you for your support!**
> > >
> > > Dear Reviewer munj,
> > >
> > > Thank you for your suggestions on our work during the review process, and also for your continued support of our work!
> > >
> > > Best regards,
> > >
> > > Authors of Submission 7851

---

### Decision · Program_Chairs · 2025-09-17

**Decision:**

Accept (poster)

**Comment:**

This paper develops an orthogonal projection approach to continual model merging that explicitly balances stability and plasticity, providing strong empirical results. The reviews were generally positive on the clarity of the approach, insights from the subspace analysis, and consistent performance improvements shown in the experiments. The rebuttal addressed most questions, clarifying some details of the approach, and adding comparisons and efficiency metrics. A few minor concerns seem to persist regarding the novelty over OPCM, and the linearity assumptions and ability to handle nonlinear components. The most negative reviewer seems to have focused on the practical necessity of the continual model merging setting, despite other reviewers viewing the setting as useful. The authors are encouraged to incorporate all aspects of the rebuttal and relevant parts of the discussion into their revision; additionally, I'd recommend strengthening the real-world motivation for CMM in the final version.